# Continuous Domain Generalization

**Zekun Cai**[1,4]    **Yiheng Yao**[2]    **Guangji Bai**[3]    **Renhe Jiang**[2*]
**Xuan Song**[1*]    **Ryosuke Shibasaki**[4]    **Liang Zhao**[3]
[1]Jilin University, Changchun, China    [2]The University of Tokyo, Tokyo, Japan
[3]Emory University, Atlanta, GA, USA    [4]LocationMind, Tokyo, Japan
{caizekun,songxuan}@jlu.edu.cn, yihengyao@g.ecc.u-tokyo.ac.jp
{guangji.bai,liang.zhao}@emory.edu, {jiangrh,shiba}@csis.u-tokyo.ac.jp

## Abstract

Real-world data distributions often shift continuously across multiple latent factors such as time, geography, and socioeconomic contexts. However, existing domain generalization approaches typically treat domains as discrete or as evolving along a single axis (e.g., time). This oversimplification fails to capture the complex, multidimensional nature of real-world variation. This paper introduces the task of Continuous Domain Generalization (CDG), which aims to generalize predictive models to unseen domains defined by arbitrary combinations of continuous variations. We present a principled framework grounded in geometric and algebraic theories, showing that optimal model parameters across domains lie on a low-dimensional manifold. To model this structure, we propose a Neural Lie Transport Operator (NeuralLio), which enables structure-preserving parameter transitions by enforcing geometric continuity and algebraic consistency. To handle noisy or incomplete domain variation descriptors, we introduce a gating mechanism to suppress irrelevant dimensions and a local chart-based strategy for robust generalization. Extensive experiments on synthetic and real-world datasets, including remote sensing, scientific documents, and traffic forecasting, demonstrate that our method significantly outperforms existing baselines in both generalization accuracy and robustness. Code is available at: `https://github.com/Zekun-Cai/NeuralLio`.

## 1 Introduction

Distribution shift refers to changes in data distributions between training and deployment environments, where the input-output relationship learned during training no longer holds at test time. This discrepancy fundamentally compromises the reliability of learned models and motivates the study of Domain Generalization (DG) [52; 49; 23; 14; 43], which aims to train models on multiple source domains that generalize to unseen domains without accessing target-domain data. While early DG methods treat domains as independent entities, recent work recognizes that distribution shifts over time follow evolutionary patterns and thus leverages temporal information to capture such regularities. This perspective has led to the Temporal Domain Generalization (TDG) [28; 40; 44; 2; 56; 59; 6], which models domain evolution along a temporal axis and seeks to generalize to future domains.

A foundational modeling assumption in TDG is that domain evolution can be projected onto a single latent axis, typically instantiated as time. This simplification reduces domain generalization to a model extrapolation problem, enabling the use of well-established sequence modeling techniques. For instance, state transition matrices simulate domain progression [44; 59]; recurrent neural networks capture temporal dependencies [2]; and ordinary differential equations model the continuous-time domain dynamics [6; 58]. While the temporal formulation offers tractability, reducing domain evolution to a unidimensional sequence imposes a fundamental limitation. In practice, distribution shifts rarely unfold along a single direction but are instead continuously driven by multiple interacting factors such as climate conditions, socioeconomic status, infrastructure development, and population structure, each governing one aspect of the generative process. These factors jointly define a

---

* Corresponding author

39th Conference on Neural Information Processing Systems (NeurIPS 2025).

continuous variation space, over which distribution shifts co-evolve smoothly. Collapsing such evolution onto a single temporal axis inevitably induces information loss and structural distortion.

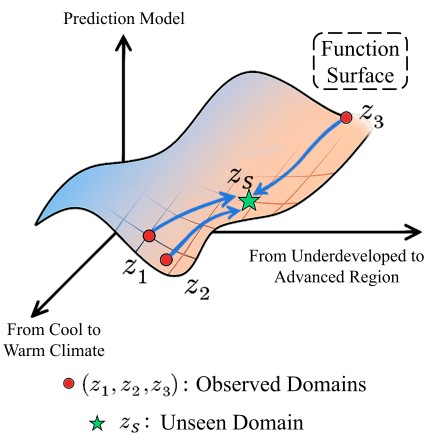

Figure 1: Illustration of continuous domain generalization. Real-world distributions are shaped by continuously varying factors. The observed domains provide sparse supervision over the joint variation-model space. Continuous domain generalization studies how predictive functions evolve over this space, enabling the model to generalize to unseen domains across the underlying continuous fields.

Formally, this paper introduces a problem termed Continuous Domain Generalization (CDG), which aims to generalize predictive models across domains that evolve over an arbitrary continuous space. This formulation captures a more general and realistic setting where domain variation arises continuously from multiple underlying factors. For instance, as shown in Fig. 1, in remote sensing, image semantics vary with seasonal climate and regional development level, which jointly shape land appearance: climate governs vegetation coverage and coloration, and development determines infrastructure density and land use. A collection of observed domains, spanning combinations such as urban areas in winter or rural areas in summer, provides supervision to calibrate predictive models across the variation space. A new region with moderate development and autumn conditions naturally resides within this space, and its corresponding predictive model should emerge through evolution along the continuous variation field, yielding predictions that are both semantically coherent and structurally grounded. This mechanism similarly governs many real-world applications. In healthcare, patient distributions may vary continuously with age, blood pressure, glucose, cholesterol, and body mass index. In urban analytics, regional traffic patterns evolve jointly with time and economic activity density. In visual recognition, image domains shift with illumination, viewpoint, and camera parameters such as focal length. All of these factors continuously alter the underlying decision function.

Continuous domain generalization introduces a novel and broadly applicable formulation. However, it poses three fundamental challenges: **1) Identifying domain variation to model evolution in general spaces.** Continuous domain generalization requires characterizing how domain variation over a high-dimensional space induces associated transitions in model behavior. However, this relationship remains theoretically unclear, with no principled constraints to ensure the coherent evolution of models across domains. **2) Modeling without inherent structural priors of domains.** Temporal domain generalization benefits from the chronological structure of domains, which enables the use of mature sequential modeling techniques. In contrast, general continuous space lacks such inherent topology, making it considerably challenging to recover the global structure of domains and to infer continuous variation fields from sparse and irregular observations. **3) Learning under imperfect representations of domain variation.** While continuous domain generalization leverages readily available contextual variables as descriptor signals to guide generalization, these proxies are often noisy, incomplete, or misaligned with the true underlying variation. Such discrepancies introduce uncertainty into the learned parameter transitions and undermine generalization performance.

To address these challenges, we propose a generic continuous domain generalization framework, which generalizes the model parameters to unseen domains using observed domains across continuous space by transport operator techniques. We theoretically show that the family of domain-optimal models constitutes a low-dimensional geometric manifold, providing a principled foundation for modeling continuous domain generalization. Building on this insight, we integrate Lie Group theory with a neural transport operator to enforce both geometric continuity and algebraic consistency. To ensure robust generalization under real-world imperfections, we incorporate a gating mechanism and a local chart-based strategy. Our framework jointly resolves the theoretical, structural, and practical challenges in continuous domain generalization. Extensive experiments on synthetic and real-world datasets, including remote sensing, scientific documents, and traffic forecasting, demonstrate that our method significantly outperforms existing baselines in both generalization accuracy and robustness.

## 2 Related Works

**Domain Generalization (DG) and Domain Adaptation (DA).** Domain Generalization (DG) aims to train models on multiple source domains that generalize to unseen ones without access to target data [39; 38; 29; 3; 13; 57]. Approaches include data augmentation [48; 49; 33; 43; 59], domain-invariant representation learning [17; 30; 19; 35], and meta-learning strategies [36; 27; 14; 7; 23]. Domain Adaptation (DA), by contrast, requires access to target data during training, using adversarial training [17; 50; 37; 51; 53] feature alignment to bridge source-target gaps [4; 22; 15; 31], and ensemble methods [47]. *Despite their effectiveness, these methods assume domains are categorical and independent, overlooking scenarios where domain shifts arise from continuous, structured variation.*

**Modeling Relationships Across Domains.** Beyond treating domains as independent categories, a line of research seeks to model relationships or structures among domains. Before deep learning, early works modeled intermediate domain shifts via feature flows, such as geodesic flow kernels [18] and manifold alignment [20]. These methods relied on handcrafted features and linear subspace, limiting their applicability to modern deep, high-dimensional representations. Temporal Domain Generalization (TDG) [28; 40; 44; 2; 59; 6] extends the idea by assuming domains follow a sequential order, enabling recurrent or extrapolative generalization. However, these methods are restricted to the temporal axis and fail to generalize to continuous domains characterized by multi-dimensional, non-sequential variation, limiting their applicability in broader settings. Other approaches leverage partial domain relationships based on domain indices. For instance, AdaGraph [37] constructs graphs over domain indices, while CIDA [51] defines domain relations through pairwise distances of domain indices. However, graphs are discrete, and distance-based formulations only induce a metric space, all of which lack the ability to express a global mapping over the domain variation space. *Overall, while these methods capture partial domain relationships, they are limited in scope and fail to capture the full structure of continuous domain variation.*

**Geometric and Algebraic Structures in Learning.** Geometric deep learning aims to uncover the low-dimensional structures and regularities underlying real-world data, formulating learning problems through unified geometric and algebraic principles [5]. It views data and models as entities living on structured spaces rather than in Euclidean coordinates, allowing learning to exploit intrinsic relationships. A broad spectrum of mathematical tools, including topology, geometry, group theory, graphs, and differential operators, provide rigorous formalisms to express physical and statistical regularities such as symmetry, invariance, equivariance, continuity, and conservation. These foundations have catalyzed major advances in scientific machine learning and physics-informed modeling [45; 21; 8; 16; 34; 24], enabling neural architectures to obey governing equations and preserve physical symmetries. Building on this foundation, we introduce a structural modeling perspective across domain-specific models for continuous domain generalization.

## 3 Problem Definition

**Continuous Domain Generalization (CDG)** CDG addresses the task of generalizing predictive models across domains whose data distributions evolve continuously with respect to underlying latent factors. Formally, let $\{\mathcal{D}_i\}_{i=1}^N$ be a collection of domains, where each domain $\mathcal{D}_i$ consists of a dataset $(X_i, Y_i)$ containing input–output pairs sampled from a distribution over $\mathcal{X} \times \mathcal{Y}$, with $\mathcal{X}$ and $\mathcal{Y}$ denoting the input and output spaces, respectively. In addition, each domain is associated with a descriptor $z_i \in \mathbb{R}^d$, with $z_i$ lying in a continuous descriptor space $\mathcal{Z}$ that captures domain-specific attributes such as spatial, temporal, demographic, economic, or environmental factors. We assume that the underlying data distribution of each domain is continuously governed by its descriptor, that is, the conditional distribution $P(Y \mid X, z_i)$ evolves smoothly over $\mathcal{Z}$ as $z_i$ varies. Accordingly, small perturbations in $z_i$, either along individual coordinates or jointly, induce proportionally small changes in the conditional distribution.

The *goal* of CDG is, during training, we are provided with a set of $N$ domains $\{\mathcal{D}_1, \mathcal{D}_2, \ldots, \mathcal{D}_N\}$, associated with their descriptors $\{z_1, z_2, \ldots, z_N\}$. For each domain $\mathcal{D}_i$, we learn a predictive model $g(\cdot; \theta_i) : \mathcal{X} \to \mathcal{Y}$, where $\theta_i \in \mathbb{R}^D$ denotes the model parameters learned from $\mathcal{D}_i$. We want to learn the co-variations between domain descriptors $\{z_1, z_2, \ldots, z_N\}$ and model parameters $\{\theta_1, \theta_2, \ldots, \theta_N\}$, and to infer the parameters $\theta_s$ corresponding to an unseen domain $\mathcal{D}_s$ given a new descriptor $z_s \notin \{z_1, \ldots, z_N\}$. This enables the learned framework to generalize to arbitrary domains across the descriptor space $\mathcal{Z}$, yielding a predictive model $g(\cdot; \theta_s)$ for any given descriptor $z_s$.

CDG assumes that data distributions evolve continuously, consistent with the continuum hypothesis in physics and engineering, which posits that macroscopic systems change smoothly with respect to their underlying causal factors. Such smooth evolution is widely observed in physical, biological, and socioeconomic systems. Consequently, modeling domain evolution as a continuous process offers a principled and tractable abstraction of real-world dynamics. Nevertheless, generalizing predictive models over such continuous spaces poses significant challenges. First, domain variation and model behavior are often entangled, making it difficult to establish stable correspondences. Second, the absence of inherent chronological organization leaves no on-hand structural guidance for modeling cross-domain relationships. Third, domain-level information is typically noisy, incomplete, or misaligned, introducing uncertainty that undermines reliable generalization. In the following sections, we tackle these challenges sequentially.

## 4 Methodology

In this section, we present our framework for continuous domain generalization through structural model transport. Specifically, we first identify the geometric structure underlying model parameters across domains, and formally prove that the collection of domain-wise parameters forms an embedded submanifold. (Section 4.1). This establishes the geometric foundation for representing model evolution as a continuous mapping on the manifold. We then derive the necessary structural constraints that should be satisfied for model evolution, and propose a Neural Lie Transport Operator (NeuralLio) grounded in Lie Group theory to enable equivariant parameter transitions across domains (Section 4.2). Finally, we handle imperfections in the descriptor space, including degeneracy and incompleteness, using gating and local chart-based modeling (Section 4.3).

### 4.1 Identifying the Parameter Manifold of Continuous Domains

This subsection provides a theoretical perspective on the geometric structure of optimal model parameters across continuous domains, revealing a smooth and coherent mapping from domain descriptors to model parameters. Preserving this structure is fundamental for continuous generalization.

**Theorem 1** (Parameter Manifold). *In continuous domain generalization, the function mapping domain descriptors to their corresponding predictive model parameters, $\theta(z) : \mathcal{Z} \to \Theta$, is continuous. Moreover, let $\mathcal{Z} \subseteq \mathbb{R}^d$ denote a theoretical descriptor space that provides a complete and non-degenerate representation of all latent factors governing domain variation, then the image set [1] $\mathcal{M} := \{\theta(z) \mid z \in \mathcal{Z}\}$ forms a $d$-dimensional embedded submanifold of $\mathbb{R}^D$.*

*Proof.* Under continuous domain distribution shift, the variation of the distribution can be described by a continuous vector field $f$. Since the optimal predictor $g(\cdot; \theta(z))$ serves as a functional representation of the conditional distribution $P(Y|X; z)$, the evolution of the predictive model function space can be modeled by:

$$\nabla_z g(\cdot; \theta(z)) = f(g(\cdot; \theta(z)), z), \tag{1}$$

where $\nabla_z$ denotes the gradient operator with respect to $z$.

Applying the chain rule to the above PDE yields:

$$\nabla_z g(\cdot; \theta(z)) = J_g\big(\theta(z)\big) \nabla_z \theta(z) \quad \implies \quad \nabla_z \theta(z) = J_g\big(\theta(z)\big)^\dagger f\big(g(\cdot; \theta(z)), z\big), \tag{2}$$

where $J_g^\dagger$ denotes the Moore–Penrose pseudoinverse of the Jacobian of $g$ with respect to $\theta$. This shows that $\theta : \mathcal{Z} \to \Theta$ is differentiable, and hence continuous.

If the descriptor space $\mathcal{Z}$ provides a complete and non-degenerate representation of all latent factors governing domain variation [2], then the Jacobian $J_z \theta(z) \in \mathbb{R}^{D \times d}$ is of full rank $d$, and the mapping $z \mapsto P(Y|X; z)$ is injective. Since the model architecture $g$ is fixed, and each domain-specific parameter $\theta(z)$ is uniquely determined by the corresponding conditional distribution $P(Y|X; z)$, the mapping $z \mapsto \theta(z)$ is also injective. Then, by the Constant Rank Theorem, the image set

$$\mathcal{M} := \{\, \theta(z) \mid z \in \mathcal{Z} \,\} \tag{3}$$

forms a $d$-dimensional embedded submanifold of $\mathbb{R}^D$.

$\square$

---

[1] The image set of a function refers to the set of all values it outputs.

[2] Such a descriptor space theoretically exists; in practice, observed descriptor serves as approximations without affecting the validity of the theorem.

Theorem 1 carries several important implications as follows:

**Property 1** (Unified Representation). *Modeling in parameter space yields a unified, comparable representation, abstracted away from task-specific feature engineering.*

**Property 2** (Geometric Regularization). *The manifold structure regularizes the parameter space by constraining optimization to a geometrically valid subset, reducing the searching complexity in $\mathbb{R}^D$.*

**Property 3** (Analytical Operability). *The manifold structure enables principled geometric and algebraic operations over model functions, facilitating interpretable and controllable generalization.*

## 4.2 Neural Domain Transport Operator under Structural Constraints

Eq. (2) reveals that a differential equation governs the evolution of model parameters. In the temporal case where the domain descriptor is one-dimensional, prior work [6] models such evolution with NeuralODEs [9]. A natural extension is to lift the ODE to a PDE framework for multi-dimensional cases. However, this introduces substantial challenges: (1) Classical numerical PDE solvers, such as finite difference or finite element methods, require known values of $\theta(z)$ on a structured grid to specify boundary or initial conditions. However, in continuous domain generalization, the $\theta(z)$ for each training domain is not directly known and must be learned through iterative optimization, leaving such conditions undefined. (2) The training domains are sparsely and irregularly distributed, precluding natural ordering or integration paths, making it infeasible to reduce the PDE to a set of ODEs along any direction. (3) Modern neural PDE solvers, such as Physics-Informed Neural Networks [45], rely on symbolic PDE formulations to define residual losses, yet the governing equations describing distributional evolution remain unknown in continuous domain generalization.

**Structural Constrained Domain Transport**   Rather than solving the PDE or explicitly regressing the parameter field $\theta(z)$, we propose to learn a structure-preserving transport operator $\mathcal{T} : \Theta \times \mathcal{Z} \times \mathcal{Z} \to \Theta$, which maps parameters at one descriptor $z_i$ to another target descriptor $z_j$. This structural operator circumvents limitations of classical and neural PDE solvers by directly learning pairwise parameter transitions. Specifically, given $\theta(z_i)$ at $z_i$ and the target point $z_j$, the operator produces

$$\theta(z_j) = \mathcal{T}(\theta(z_i),\, z_i,\, z_j). \tag{4}$$

To ensure that $\mathcal{T}$ yields meaningful and generalizable parameter transitions, we explore and formulate the necessary conditions that the transport operator needs to satisfy for continuous domain generalization: a geometric continuity structure and an algebraic group structure.

**Definition 1** (Geometric Structure). *The neural transport operator $\mathcal{T}$ is said to satisfy geometric structure if it is continuous in all of its inputs.*

**Definition 2** (Algebraic Structure). *The neural transport operator $\mathcal{T}$ is said to satisfy algebraic structure if the following properties hold:*

- **Closure:** $\mathcal{T}(\theta(z_i), z_i, z_j) \in \Theta$, *ensuring transported parameters remain within the valid space.*
- **Identity:** $\mathcal{T}(\theta(z_i), z_i, z_i) = \theta(z_i)$, *ensuring that self-transport leaves parameters unchanged.*
- **Associativity:** $\mathcal{T}(\mathcal{T}(\theta(z_i), z_i, z_j), z_j, z_k) = \mathcal{T}(\theta(z_i), z_i, z_k)$, *ensuring that sequential transports are equivalent to direct ones.*
- **Invertibility:** $\mathcal{T}^{-1}\mathcal{T}(\theta(z_i), z_i, z_j) = \theta(z_i)$, *ensuring that each transport can be exactly reversed.*

The geometric structure ensures that $\theta(z)$ forms a smooth manifold, as established in Theorem 1. The algebraic structure defined via *Group*-like axioms imposes *Equivariance*. Equivariance characterizes a symmetry relation between transformations in the input and output spaces. In this context, it ensures that applying a transformation in the descriptor space and then mapping to parameters yields the same result as transforming parameters directly. Without equivariance, the learned operator may produce parameter trajectories that deviate from the true evolution of the underlying data distribution.

**Neural Lie Transport Operator**   The geometric and algebraic structure jointly imply that the parameter family $\mathcal{M} = \{\theta(z) \mid z \in \mathcal{Z}\}$ forms *Lie Group*—i.e., a smooth manifold with a compatible group operation. The transport operator induces parameter Lie Group transitions along directions specified by domain descriptors. This motivates us to propose the **Neural Lie Transport Operator (NeuralLio)**, a learnable operator grounded in Lie theory and parameterized by a neural network. NeuralLio characterizes the local variation of $\theta(z)$ using the Lie algebra $\mathfrak{g}$ generated from the domain

descriptor. The Lie algebra $\mathfrak{g}$ serves as a tangent space that captures the differential structure of the parameter manifold. Given a source descriptor $z_i$ and corresponding parameter $\theta(z_i)$, we generate a Lie algebra element $V(z_i) \in \mathfrak{g}$ using a neural network. For a target descriptor $z_j$, we apply the offset $(z_j - z_i)$ to $V(z_i)$ to transport $\theta(z_i)$ to $\theta(z_j)$:

$$\theta(z_j) = \mathcal{T}\big(\theta(z_i), z_i, z_j\big) = \exp\big((z_j - z_i)V(z_i)\big) \cdot \theta(z_i), \tag{5}$$

where the exponential map $\exp : \mathfrak{g} \to \mathcal{M}$ lifts the Lie algebra vector to a valid group transformation. In practice, the parameter vector $\theta(z) \in \mathbb{R}^D$ is high-dimensional. We first encode it into a compact latent representation $e(z) = f_e(\theta(z)) \in \mathbb{R}^m$ using an encoder $f_e$. We then perform cascaded transport in the latent space. Specifically, we define a set of $d$ neural Lie algebra fields $\{V^1, \ldots, V^d\}$, where each $V^k \in \mathbb{R}^{m \times m}$ is produced by a field network $f_v^k : \mathbb{R}^d \to \mathbb{R}^{m \times m}$ conditioned on the source descriptor $z_i$. Each $f_v^k$ can be instantiated with a generic deep neural network. Given a descriptor shift $(z_j - z_i)$, we compute the cumulative transformation as:

$$e(z_j) = \left( \prod_{k=1}^{d} \exp\left( (z_j^k - z_i^k) \cdot f_v^k(z_i) \right) \right) \cdot e(z_i). \tag{6}$$

The latent representation $e(z_j)$ is then decoded by $f_d : \mathbb{R}^m \to \mathbb{R}^D$ to obtain the final $\theta(z_j)$.

### 4.3 Handling Imperfect Descriptors Space

Theorem 1 discusses a theoretically grounded descriptor space. In the real world, however, the descriptors accessible for model training are observable projections of this theoretical space. They may contain noise, redundancy, or missing factors, leading to imperfect representations. These imperfections introduce two challenges: (1) Degeneracy: Some directions in the observed $\mathcal{Z}$ fail to affect $\theta(z)$ or redundantly encode variations. (2) Incompleteness: The observed descriptors partially capture the degrees of freedom governing domain shifts, leaving some latent factors unmodeled.

**Suppressing Degeneracy via Descriptor Gating**  We introduce a gating mechanism to suppress irrelevant or redundant directions in $\mathcal{Z}$. Specifically, we apply a dimension-wise gate to modulate the influence of each feature in $z$, enabling the model to focus on informative directions for parameter variation. The gate consists of two components: a data-dependent gate defined as $\mathbf{g}(z_i) = \text{Sigmoid}(W z_i)$ with $W \in \mathbb{R}^{d \times d}$, and a global trainable gate vector $\mathbf{w} \in \mathbb{R}^d$ shared across all domains. This yields a shared gating vector $\mathbf{m}(z_i) = \mathbf{g}(z_i) \odot \mathbf{w}$ used to modulate both source and target descriptors:

$$\tilde{z}_i = z_i \odot \mathbf{m}(z_i), \quad \tilde{z}_j = z_j \odot \mathbf{m}(z_i). \tag{7}$$

where $\odot$ denotes element-wise multiplication. This gating suppresses degenerate directions before computing the transport operator.

**Mitigating Incompleteness via Local Chart**  When the space $\mathcal{Z}$ is incomplete, learning a globally consistent transport function becomes underdetermined. We adopt a localized modeling strategy grounded in the theory of differential geometry: the parameter manifold is represented as an *atlas*, i.e., a collection of overlapping local *charts*. For each descriptor $z_i$, we define its neighborhood as:

$$\mathcal{N}(z_i) := \{z_j \in \mathcal{Z} \mid z_j \text{ is a } k\text{-NN of } z_i\}, \tag{8}$$

and restrict the transport operator to locally adjacent domains:

$$\theta(z_j) = \mathcal{T}(\theta(z_i), z_i, z_j), \quad \forall z_j \in \mathcal{N}(z_i). \tag{9}$$

The localized construction enables the mapping $z \mapsto \theta(z)$ to remain smooth within each chart, while the union of charts compensates for global non-smoothness from incompleteness.

**Optimization**  NeuralLio is optimized by supervising the transport operator $\mathcal{T}$ over local neighborhoods in the descriptor space. For each training descriptor $z_i$, we sample nearby descriptors $z_j \in \mathcal{N}(z_i)$ and train $\mathcal{T}$ to transport model parameters from $z_i$ to $z_j$ so that the resulting predictions and latent representations remain consistent. During inference, parameters for an unseen domain $z_s$ are inferred by transporting from the nearest training descriptor using the learned operator. Detailed procedures for both training and inference are summarized in Appendix B. We also provide a detailed model complexity analysis in Appendix A.4.

Table 1: Performance comparison on continuous domain datasets. Classification tasks report error rates (%) and regression tasks report MAE. 'N/A' implies that the method does not support the task.

| Model | Classification | | | | | Regression |
|---|---|---|---|---|---|---|
| | 2-Moons | MNIST | fMoW | ArXiv | Yearbook | Traffic |
| Descriptor-Agnostic | | | | | | |
| ERM | $34.7 \pm 0.2$ | $31.8 \pm 0.9$ | $27.7 \pm 1.6$ | $35.6 \pm 0.1$ | $8.6 \pm 1.0$ | $16.4 \pm 0.1$ |
| IRM [1] | $34.4 \pm 0.2$ | $33.0 \pm 0.8$ | $41.5 \pm 2.8$ | $37.4 \pm 1.0$ | $8.3 \pm 0.5$ | $16.6 \pm 0.1$ |
| V-REx [26] | $34.9 \pm 0.1$ | $32.2 \pm 1.4$ | $32.1 \pm 3.6$ | $37.3 \pm 0.7$ | $8.9 \pm 0.5$ | $20.9 \pm 0.6$ |
| GroupDRO [46] | $34.5 \pm 0.1$ | $37.6 \pm 1.0$ | $28.6 \pm 1.9$ | $35.6 \pm 0.1$ | $8.0 \pm 0.4$ | $16.2 \pm 0.1$ |
| Mixup [54] | $34.9 \pm 0.1$ | $34.0 \pm 0.9$ | $27.1 \pm 1.5$ | $35.5 \pm 0.2$ | $7.5 \pm 0.5$ | $16.1 \pm 0.1$ |
| DANN [17] | $35.1 \pm 0.4$ | $34.7 \pm 0.6$ | $\underline{26.0 \pm 0.7}$ | $36.5 \pm 0.2$ | $8.9 \pm 1.4$ | $18.1 \pm 0.2$ |
| MLDG [27] | $34.6 \pm 0.2$ | $85.1 \pm 2.5$ | $29.2 \pm 1.0$ | $35.8 \pm 0.2$ | $7.7 \pm 0.5$ | $16.9 \pm 0.1$ |
| CDANN [32] | $35.0 \pm 0.2$ | $36.4 \pm 0.8$ | $27.6 \pm 0.9$ | $36.2 \pm 0.2$ | $8.7 \pm 0.4$ | $17.3 \pm 0.2$ |
| URM [25] | $34.7 \pm 0.1$ | $31.8 \pm 1.3$ | $26.9 \pm 1.0$ | $35.5 \pm 0.4$ | $8.0 \pm 0.3$ | $16.2 \pm 0.2$ |
| Descriptor-Aware | | | | | | |
| ERM-D | $\underline{13.1 \pm 1.5}$ | $31.7 \pm 0.5$ | $28.9 \pm 1.8$ | $38.1 \pm 0.6$ | $7.4 \pm 0.5$ | $\underline{15.9 \pm 0.1}$ |
| NDA | $25.4 \pm 0.3$ | $\underline{26.3 \pm 0.7}$ | $31.2 \pm 1.4$ | $35.6 \pm 0.6$ | $11.0 \pm 0.8$ | $17.2 \pm 0.2$ |
| CIDA [51] | $14.2 \pm 1.1$ | $27.4 \pm 0.5$ | $27.1 \pm 0.9$ | $\underline{35.3 \pm 0.4}$ | $8.4 \pm 0.8$ | $16.6 \pm 0.1$ |
| TKNets [59] | N/A | N/A | N/A | N/A | $8.4 \pm 0.3$ | N/A |
| DRAIN [2] | N/A | N/A | N/A | N/A | $10.5 \pm 1.0$ | N/A |
| Koodos [6] | N/A | N/A | N/A | N/A | $\underline{6.6 \pm 1.3}$ | N/A |
| **NeuralLio (Ours)** | $\mathbf{3.2 \pm 1.2}$ | $\mathbf{9.5 \pm 1.1}$ | $\mathbf{24.5 \pm 0.5}$ | $\mathbf{34.7 \pm 0.4}$ | $\mathbf{4.8 \pm 0.3}$ | $\mathbf{15.1 \pm 0.1}$ |

## 5 Experiment

In this section, we evaluate the effectiveness of the proposed framework across diverse continuous domain generalization tasks. Our experimental study is designed to answer the following key questions: *1) Can the framework effectively generalize predictive models across continuously evolving domains? 2) Can the framework faithfully recover the underlying parameter manifold shaped by domain variation? 3) Can the framework robustly handle descriptor imperfections? 4) Can the learned transport operator exhibit the desired structural properties?* More detailed results (i.e., dataset details, baseline details, model and hyperparameter configurations, ablation study, scalability analysis, sensitivity analysis, and convergence analysis) are demonstrated in Appendix A.

**Synthetic Datasets.** Two synthetic datasets are employed to simulate continuous domain shifts under interpretable variations. In the 2-Moons dataset, each domain is generated by applying scaling and rotation to the base moon shape. The descriptor $z = [z^1, z^2] \in \mathbb{R}^2$ encodes the scale factor and rotation angle, respectively. We train the model on 50 randomly sampled domains, and evaluate it on 150 additional randomly sampled domains, together with extra test domains uniformly sampled over a mesh grid in the descriptor space (see Fig. 5). In the MNIST dataset, each domain is constructed by applying rotation, color shift (red to blue), and occlusion to digits. The descriptor $z = [z^1, z^2, z^3] \in \mathbb{R}^3$ encodes the intensity of each transformation. We use 50 randomly selected domains for training and another 50 for testing. Visual illustrations are shown in Fig. 6.

**Real-world Datasets.** We further evaluate our framework on a diverse collection of multimodal real-world datasets: fMoW, ArXiv, and Yearbook for classification, and Traffic for regression. For each dataset, we construct continuous descriptors using auxiliary information derived from publicly available domain metadata or contextual variables. Comprehensive details of dataset composition, processing, and descriptor construction are provided in Appendix A.1.

**Baselines.** We employ two categories of baselines. The *Descriptor-Agnostic* group operates purely on input–output pairs without domain-level descriptor: ERM, IRM, V-REx, GroupDRO, Mixup, DANN, MLDG, CDANN, and URM. The *Descriptor-Aware* group incorporates explicit domain descriptors to guide generalization: ERM-D, NDA, CIDA[51], TKNets[59], DRAIN[2], and Koodos[6]. Details of all comparison methods are provided in Appendix A.2.

**Metrics.** Error rate (%) is used for classification tasks. Mean Absolute Error (MAE) is used for regression tasks. All models are trained on training domains and evaluated on all unseen test domains. Each experiment is repeated five times, and we report the mean and standard deviation. Full hyperparameter settings and implementation details are provided in Appendix A.3.

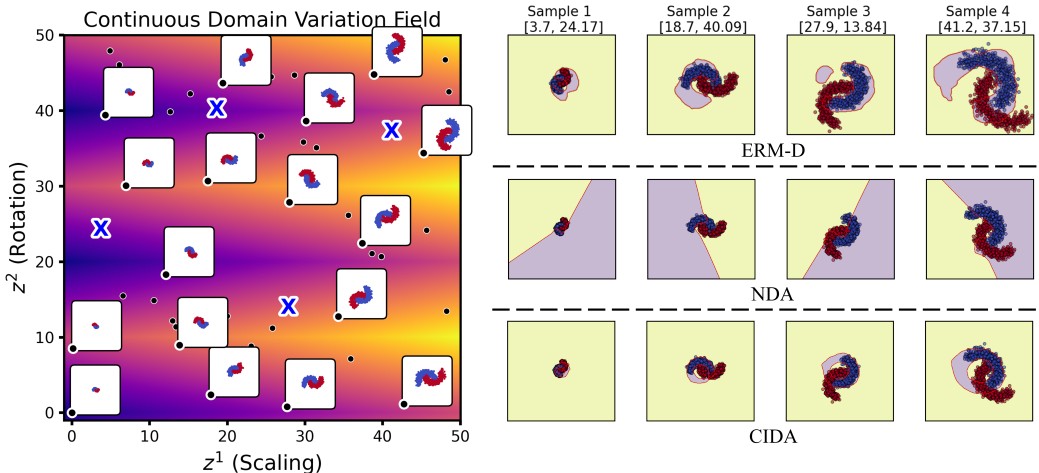

Figure 2: Visualization of generalization behavior of baseline models on the 2-Moons dataset. **Left:** All training domains (black dots) and selected test domains (blue crosses) in the variation space. **Right:** Decision boundaries of baseline methods (rows) evaluated at the four test domains (columns).

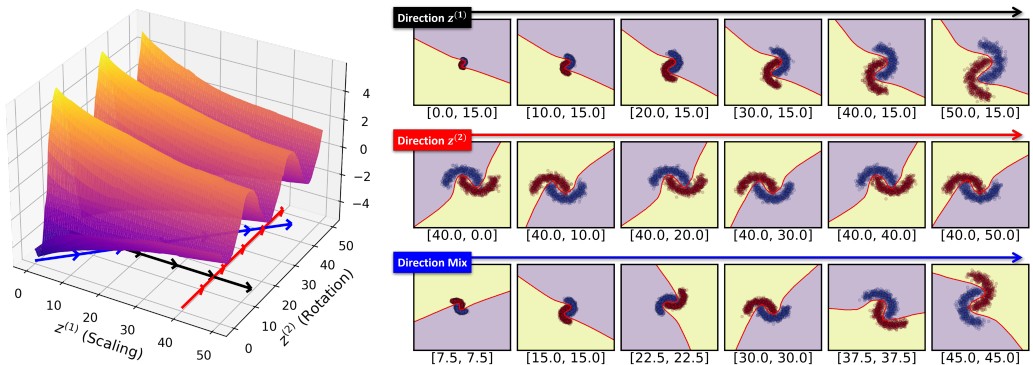

Figure 3: Visualization of the learned parameter manifold and the corresponding generalization behavior. **Left:** PCA projection of predicted parameters $\theta(z)$ over the entire descriptor space. **Right:** Visualization of decision boundaries and data samples along selected direction.

## 5.1 Quantitative Analysis

We present the performance of our proposed method against baseline methods, highlighting results from Table 1. Our method demonstrates strong generalizability across a wide range of continuous domains, with particularly large improvements on 2-Moons (3.2% vs. 13.1%) and MNIST (9.5% vs. 26.3%), indicating its effectiveness in modeling continuous domain variation. Beyond synthetic data, NeuralLio also achieves the best results on fMoW, ArXiv, Yearbook, and Traffic, indicating its robustness in handling diverse and irregular real-world domain variation. Table 1 also reveals several insights into how existing methods behave: (1) Descriptor-aware methods generally outperform descriptor-agnostic ones, yet naïvely incorporating descriptors (e.g., ERM-D) risks spurious correlations that hinder generalization. (2) Temporal methods are insufficient for general continuous domains. Models designed for temporal domain generalization (i.e., TKNets, DRAIN, Koodos) rely on domain sequence and fail when the descriptor space lacks such an ordering structure. (3) Continuous domain generalization is intrinsically challenging. Even among descriptor-aware methods, the performance of baselines on synthetic datasets is unsatisfactory. No existing method achieves reasonable accuracy. This highlights that prior methods largely overlook the underlying structure of domain variation, which is essential for modeling continuous generalization.

## 5.2 Qualitative Analysis

To further understand how domain variation influences model behavior, we visualize both baseline and learned models on the 2-Moons dataset. Fig. 2 left provides a global view of the descriptor space,

where the two axes correspond to scaling (monotonic variation) and rotation (periodic variation). The background color encodes the continuous variation field across domains. All training domains are shown as black dots, with several illustrated in small inset panels. Four unseen test domains are randomly sampled for visualization and marked as blue crosses in the descriptor space. The right panel visualizes the classification boundaries of three baseline methods evaluated on the four test domains. Each row corresponds to a method, and each column to a test domain. ERM-D embeds descriptors directly into the model input, leading to entangled representations. This often induces spurious correlations and overfitting, resulting in highly irregular and fragmented decision boundaries. NDA relies solely on chance overlap between training and test domains, lacking any generalization mechanism. As a result, it produces unstable decision boundaries with no guarantee. CIDA performs adversarial alignment based on descriptor distances and partially captures the scaling effect, visible as a ring-like expansion. However, it fails to model rotational variation because its metric-driven objective lacks structural fidelity. This limitation highlights the inherent weakness of distance-based objectives in preserving domain geometry, particularly under non-isometric transformations.

Fig. 3 shows an intrinsic view of the parameter manifold learned by NeuralLio on the 2-Moons dataset. Specifically, we train the model using the given training domains, and then densely generalize $\theta(z)$ across the full descriptor space. The left panel visualizes the parameter manifold obtained by projecting all the predicted model parameters $\theta(z)$ via PCA. The resulting "twisted surface" aligns with the monotonic–periodic geometry of scaling and rotation, respectively. The right panel demonstrates that traversing along different descriptor directions corresponds to smooth, consistent transformations in decision boundaries, showing that the learned operator preserves geometric continuity and captures the co-evolution of model parameters and domain distributions. In particular, the mixed-direction case reveals the model's ability to jointly encode multiple domain variations. In summary, the visualization results firmly demonstrate that our method captures the structural geometry of domain variation and enables interpretable generalization across the descriptor space.

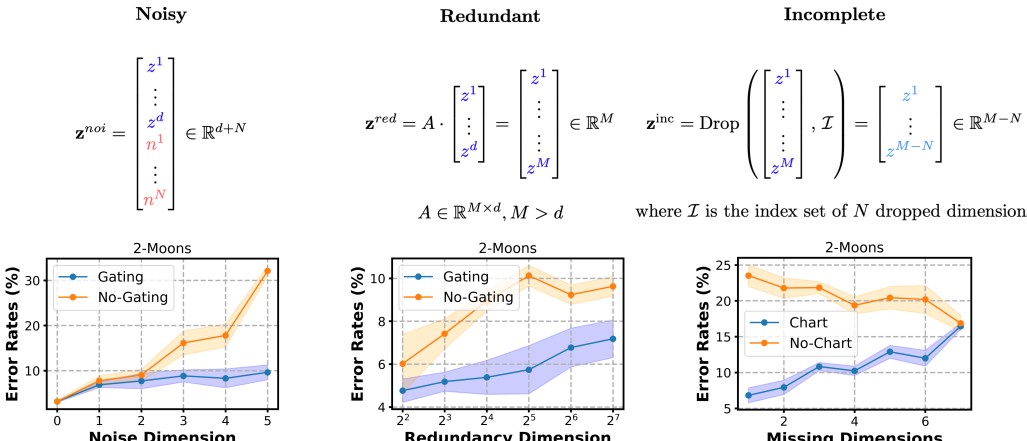

Figure 4: Robustness to imperfect descriptors. **Top:** Visualization of noisy, redundant, and incomplete descriptor constructions. **Bottom:** Error rates under increasing imperfection on the 2-Moons dataset.

### 5.3 Imperfections Analysis

We design three controlled experiments to simulate the common imperfections of the domain descriptor: noise, redundancy, and incompleteness, as discussed in Section 4.3. To assess the noise, we augment $z$ with randomly sampled noise dimensions, forming descriptors with uninformative components. To assess redundancy, we apply redundant projections to the original descriptor to obtain a higher-dimensional, over-parameterized version. This setup mimics real-world cases where descriptors are overly entangled. To study incompleteness, we simulate missing information by randomly dropping $n$ dimensions from the redundant descriptor. As $n$ increases, the retained dimensions provide less faithful representation of the underlying domain variation. An overview of the construction is illustrated in the top row of Fig. 4.

The bottom row of Fig. 4 shows the error rates under increasing levels of descriptor imperfection across three settings. In the *Noisy* setting (left), we gradually append up to five random noise

dimensions to the original descriptor. As noise increases, the performance of the model without gating rapidly deteriorates, reaching over 30% error at the highest noise level. In contrast, the model equipped with the gating mechanism remains significantly more stable. This indicates that although the noise dimensions are uninformative, the remaining structure in the descriptor space still guides transport reasonably well, provided that irrelevant dimensions are actively suppressed. In the *Redundant* setting (middle), we expand the descriptor by applying random linear projections to the original descriptor, introducing redundancy without discarding any information. Both models maintain reasonably low error rates (under 10%), since the underlying domain variation remains fully preserved. However, the model equipped with gating achieves better and more stable performance. This suggests that while redundancy itself is not harmful, the gating mechanism helps by softly recovering the original independent factors of variation, effectively regularizing the projection space and improving robustness. In the *Incomplete* setting (right), we construct an 8-dimensional descriptor space embedding two latent factors of variation, and simulate increasing incompleteness by progressively removing 1 to 7 dimensions. The model without charting (No-Chart) fails once key structural information is removed. In contrast, the chart-based variant exhibits a controlled performance decline. This validates the benefit of local manifold modeling, as restricting transport computation to descriptor neighborhoods allows the chart-based approach to preserve continuity even when global descriptor integrity is compromised.

## 5.4 Structure Property Analysis

To validate the structure properties introduced in Section 4.2, we evaluate the learned operator on the 2-Moons dataset. We measure the consistency between theoretically equivalent transformations by computing the cosine similarity of their resulting parameters across all test domains.

**Identity.** For each test descriptor $z_i$, we apply the self-transport $\theta(z_i) \xrightarrow{z_i \to z_i} \theta'(z_i)$ and compute the cosine similarity: $\cos(\theta(z_i), \theta'(z_i))$.

**Associativity.** We consider all triplet combinations $(z_i, z_j, z_k)$ from the test descriptors. For each triplet, we compute two transports: $\theta(z_i) \xrightarrow{z_i \to z_j} \theta'(z_j) \xrightarrow{z_j \to z_k} \theta'(z_k)$, and $\theta(z_i) \xrightarrow{z_i \to z_k} \theta''(z_k)$, then evaluate $\cos(\theta'(z_k), \theta''(z_k))$.

Table 2: Empirical verification of the structure properties of the learned operator.

| Property | Cosine Similarity (%) |
| --- | --- |
| Identity | 99.9 |
| Associativity | 99.1 |
| Invertibility | 98.3 |

**Invertibility.** We consider all pairs $(z_i, z_j)$ from the test descriptors. For each pair, we apply the round-trip transport $\theta(z_i) \xrightarrow{z_i \to z_j} \theta'(z_j) \xrightarrow{z_j \to z_i} \theta'(z_i)$, and compute $\cos(\theta(z_i), \theta'(z_i))$.

**Closure.** This property is inherently satisfied, as the operator always maps within the $\mathbb{R}^D$.

As shown in Table 2, the learned operator fulfills the expected properties with high precision, which empirically confirms that NeuralLio realizes the Lie-Group-based transport formulation, providing a principled foundation for structurally consistent model evolution.

## 6 Conclusion

This paper presents a unified framework for continuous domain generalization, which aligns structural model parameterization with domain-wise continuous variation. We identify the geometric and algebraic foundations for continuous model evolution and instantiate them through a neural Lie transport operator, which enforces structure-preserving parameter transitions over the variation space. Beyond idealized settings, we further address realistic imperfections in the descriptor space by introducing a gating mechanism to suppress noise, and a local chart construction to handle partial information. We provide both theoretical analysis and extensive empirical validation on synthetic and real-world datasets, demonstrating strong generalization performance, robustness, and scalability.

Building upon these insights, several directions emerge beyond the current setting. First, many seemingly discrete domain generalization tasks may in fact represent sparse samples of an underlying continuous process governed by latent semantics. Discovering these latent descriptors through unsupervised or generative modeling is appealing. Second, scaling the framework toward more expressive model structures offers a promising avenue for extending its applicability to broader domains. Third, exploring arithmetic operations between models in the functional space may further deepen the understanding of continuous generalization.

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

# A Experimental Details

## A.1 Dataset Details

To establish a comprehensive evaluation protocol, we construct a suite of synthetic and real-world datasets designed to cover diverse transformation types, domain structures, data fields, and prediction tasks. The synthetic datasets support controlled analysis of continuous distribution shifts through interpretable transformations. For real-world data, we collect or reorganize large-scale datasets across four domains in vision, language, and spatiotemporal modalities. Each domain is anchored to a real-world geography, time, semantics, or economic context.

**Synthetic Datasets** Prior temporal domain generalization studies [51; 40; 2; 59; 6] adopt single-factor domain variations. We instead introduce multiple co-occurring transformations into their widely adopted synthetic dataset, whose combinations induce complex distribution shifts and substantially increase the difficulty of generalization.

1. **2-Moons** This dataset is a variant of the classic 2-entangled moons dataset, where the lower and upper moon-shaped clusters are labeled 0 and 1, and each contains 500 instances. We construct two types of transformations jointly: scaling, which enlarges the spatial extent of the moons by 10% per unit, and rotation, which rotates the entire structure counter-clockwise by $18°$ per unit. These transformations induce a two-dimensional descriptor $z = [z^{\text{scal}}, z^{\text{rot}}] \in \mathbb{R}^2$. We train on 50 randomly sampled domains, and test on 150 randomly sampled others plus a fixed mesh grid of descriptor points. Examples from the train set are shown in Fig. 2, and the distribution of train and test domains is visualized in Fig. 5. The continuous domain shift arises from smooth transformations of the moon-shaped clusters.

2. **MNIST** This dataset is a variant of the classic MNIST dataset [12], where each domain consists of 1,000 digit images randomly sampled from the original MNIST. A combination of three transformations is applied to all samples within each domain: rotation ($18°$ counterclockwise per unit), color shift (from red to blue over the transformation space), and occlusion (increasing coverage ratio per unit). These transformations induce a three-dimensional domain descriptor $z = [z^{\text{rot}}, z^{\text{col}}, z^{\text{occ}}] \in \mathbb{R}^3$. We randomly sample 50 domains for training and another 50 for testing. Fig. 6 illustrates representative training and test domains. The visual appearance of digits exhibits substantial variation due to combinations of transformations, posing a significant challenge for models to generalize.

**Real-world Datasets**

3. **fMoW** The fMoW dataset [10] consists of over 1 million high-resolution satellite images collected globally between 2002 and 2018. Each image is labeled with the functional purpose of the region, such as airport, aquaculture, or crop fields. We select ten common categories to construct a multi-class classification task and define each country as a separate domain, resulting in 96 domains. For each country, we collect publicly available climate statistics from the CRU climate dataset[3], including seasonal-term averages of temperature, precipitation, humidity, and solar radiation. These climate factors influence land use decisions and introduce meaningful distribution shifts driven by environmental and geographic variability. We randomly select 50 domains for training and use the remaining for testing.

4. **Arxiv** The arXiv dataset [11] comprises 1.5 million pre-prints over 28 years, spanning fields such as physics, mathematics, and computer science. We construct a title-based classification task to predict the paper's subject. Domains are defined by the publisher that eventually accepted the article, yielding 83 domains. This captures semantic variation across venues; for example, the term "neural" refers to artificial networks in IEEE, but to biological systems in Nature Neuroscience. Publisher metadata is obtained from the open-source OpenAlex [42]. We randomly select 40 domains for training and use the remaining for testing.

5. **YearBook** The Yearbook dataset [55] contains frontal portraits from U.S. high school yearbooks spanning 1930 to 2013. The task is to classify gender from facial images. Same as the setting in [6], we sample 40 years from the 84-year range, treating each year as a separate domain. The resulting domains are temporally ordered. The first 28 domains are

---

[3]https://crudata.uea.ac.uk/cru/data/hrg/

used for training and the remaining for testing. This dataset serves as a one-dimensional temporal testbed, where time is treated as a special case of continuous domain generalization. It enables existing temporal domain generalization methods to be comparable with ours.

6. **Traffic** We use a real-world taxi flow dataset [41] collected from Beijing, covering the period from February to June 2015. The city is partitioned into 1,024 ($32 \times 32$) regions, and the task is time-series forecasting of future traffic flow based on past hourly taxi inflow and outflow observations. Each region is associated with the distribution of Points of Interest (POIs). We treat each region as a separate domain and use the POIs distribution as its descriptor, as it reflects functional differences in land use that potentially influence local traffic dynamics. We select 100 domains for training and use the remaining for testing.

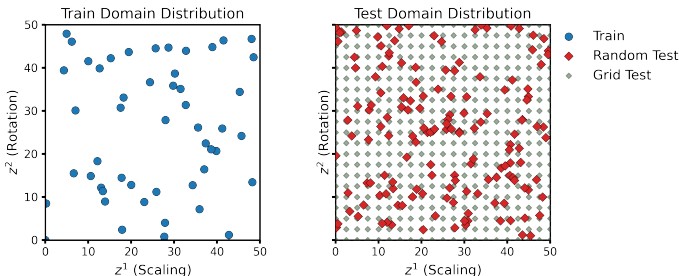

Figure 5: Train and test domain descriptors for the 2-Moons dataset. The left plot shows the 50 training domains, while the right plot shows the 150 randomly sampled test domains and additional test domains uniformly distributed over a fixed mesh grid in the descriptor space.

## A.2 Comparison Methods

**Descriptor-Agnostic Methods.** These methods operate purely on input–output pairs and do not utilize domain-level descriptor information:

- ERM (Empirical Risk Minimization): Trains a single model across all domains by minimizing average training loss.
- IRM [1]: Encourages invariant predictive relationships across environments by penalizing gradient dependence between input and loss.
- V-REx [26]: Enforces risk extrapolation to approximate invariant predictors, offering robustness to covariate shift and enabling partial causal identification.
- GroupDRO [46]: Adopts a worst-case optimization over domain-specific risks to ensure robustness.
- Mixup [54]: Applies data interpolation across domains to promote smoothness and generalization for unsupervised domain adaptation.
- DANN [17]: Introduces domain adversarial training to align latent representations across domains.
- MLDG [27]: Performs meta-learning by simulating domain shifts during training via meta-train/meta-test splits.
- CDANN [32]: An adversarial method that conditions on domain labels to enforce domain-invariant features.
- URM [25]: A recent invariant learning method that enforces uniformity across both data distributions and feature representations.

**Descriptor-Aware Methods.** These methods incorporate explicit domain descriptors to guide model generalization across domains:

- ERM-D: An extension of Empirical Risk Minimization that incorporates the domain descriptor as an auxiliary input. Specifically, the descriptor is first encoded via a shallow MLP and then concatenated with the feature representation of the input data. This allows the network to condition its prediction on domain-specific signals.
- NDA (Nearest Domain Adaptation): This method first trains a global model on all training domains, then fine-tunes it individually for each domain. For test-time prediction, it selects the closest training domain in descriptor space and directly uses its fine-tuned model.

Domain 3: 1 [16.6, 22.60, 7.65]   Domain 4: 0 [9.0, 15.12, 6.13]   Domain 8: 2 [49.7, 40.92, 12.00]   Domain 12: 3 [32.0, 13.69, 10.57]   Domain 15: 6 [4.1, 8.38, 14.52]   Domain 16: 9 [26.8, 24.72, 12.61]

Domain 19: 3 [37.7, 8.82, 12.54]   Domain 20: 9 [13.9, 29.05, 13.35]   Domain 24: 0 [30.2, 41.80, 6.98]   Domain 28: 1 [14.2, 8.20, 10.34]   Domain 33: 3 [32.0, 0.15, 8.36]   Domain 34: 7 [17.4, 21.87, 7.36]

Domain 36: 1 [3.6, 48.28, 8.72]   Domain 37: 3 [3.6, 2.75, 12.26]   Domain 39: 8 [5.6, 14.05, 14.40]   Domain 44: 0 [19.0, 2.23, 14.80]   Domain 45: 7 [4.7, 4.22, 7.60]   Domain 49: 3 [39.8, 29.09, 9.89]

**Test Domains**
**18 of 50**

Domain 50: 4 [34.5, 26.39, 9.62]   Domain 51: 2 [27.2, 40.49, 13.34]   Domain 56: 3 [3.9, 36.37, 7.69]   Domain 57: 8 [41.8, 14.48, 12.85]   Domain 61: 6 [25.5, 44.46, 6.93]   Domain 62: 3 [25.1, 39.11, 14.60]

Domain 73: 5 [1.5, 26.06, 12.96]   Domain 78: 5 [37.3, 49.75, 13.39]   Domain 84: 1 [13.6, 11.73, 8.04]   Domain 89: 6 [21.5, 10.81, 8.11]   Domain 90: 1 [22.6, 26.28, 10.39]   Domain 91: 6 [19.9, 47.41, 8.67]

Domain 93: 7 [41.1, 38.85, 7.41]   Domain 94: 1 [10.7, 7.66, 6.81]   Domain 95: 3 [30.3, 46.23, 5.55]   Domain 96: 5 [37.7, 0.22, 13.89]   Domain 97: 6 [36.9, 47.45, 13.26]   Domain 99: 5 [9.7, 41.15, 13.20]

Figure 6: Visualization of the MNIST dataset. Each domain corresponds to a unique combination of three transformations: rotation, color shift, and occlusion. One representative image is sampled from each domain for visualization. The title above each image indicates its domain index and ground-truth label, while the descriptor is shown below. The top and bottom panels respectively show 18 randomly sampled training and test domains.

- CIDA [51]: CIDA addresses domain adaptation for continuously indexed domains. It employs a descriptor-conditioned discriminator that predicts the domain index value from encoded features. The encoder is trained adversarially to obfuscate this prediction, encouraging domain-invariant representations across domains. In our setup, we adopt the multi-dimensional extension of CIDA, adapting the discriminator to regress continuous descriptors.

- TKNets [59]: Learns the temporal evolution of the model by mapping them into a Koopman operator-driven latent space, where the dynamics are assumed to be linear. Each temporal domain has a shared latent linear operator that governs its transitions.

- DRAIN [2]: Models the temporal evolution of predictive models under distributional drift using a recurrent neural architecture. Specifically, DRAIN employs an LSTM-based controller to dynamically generate model parameters conditioned on temporal dependency, enabling the model to adapt to continuously shifting data distributions without access to future domains.

- Koodos [6]: Models the continuous evolution of model parameters with neural ODEs, where the dynamics are driven by temporal dependency. It synchronizes the progression of data distributions and model parameters through a latent learnable dynamics.

## A.3 Model Configuration

We provide full implementation details and code in our repository[4]. Our model comprises four jointly trained components: a per-domain predictive model, a parameter encoder-decoder pair, a neural transport operator, and a gating mechanism. We adopt a task-specific predictive architecture (detailed below) shared across all domains. Each training domain is associated with its own parameter vector, while maintaining structural consistency. The parameters encoder and decoder are four-layer MLPs with ReLU activations. The gating mechanism consists of a linear transformation and a learnable mask. The neural transport operator is implemented as an exponentiated vector field over the descriptor space. It includes two sub-networks: a *field network* that maps the source descriptor $z_i$ to a set of basis matrices, and a *coefficient network* that maps the descriptor difference $\Delta z = z_j - z_i$ to a set of scalar weights. The resulting transformation is computed by sequentially applying matrix exponentials of the scaled basis matrices as described in Eq. 6. For models with large parameter sizes, we introduce a shared feature extractor and infer only the remaining domain-specific parameters. All experiments are conducted on a 64-bit machine with two 20-core Intel Xeon Silver 4210R CPUs @ 2.40GHz, 378GB memory, and four NVIDIA GeForce RTX 3090 GPUs. Results are averaged over five runs with different random seeds using the Adam optimizer. We specify the task-specific predictive architecture for each dataset as follows:

1. **2-Moons** The predictive model is a three-layer MLP with 50 hidden units per layer and ReLU activations. The encoder and decoder are both four-layer MLPs with layer dimensions $[1024, 512, 128, 32]$. The transport operator consists of a 32-dimensional linear field network with 2 basis matrices. The learning rate is set to $1 \times 10^{-3}$.

2. **MNIST** The shared feature extractor is a convolutional backbone composed of three convolutional layers with channels $[32, 32, 64]$, each followed by a ReLU activation and a max pooling layer with kernel size 2. The resulting features are flattened and passed through a dropout layer. The per-domain predictive model is a two-layer MLP with a hidden dimension of 128 and an output dimension of 10. The encoder and decoder are four-layer MLPs with layer dimensions $[1024, 512, 128, 32]$. The neural transport operator is implemented as a 32-dimensional linear field network with 3 basis matrices. The learning rate is set to $1 \times 10^{-3}$ for all components.

3. **fMoW** ResNet-50 backbone pretrained on ImageNet as the shared feature extractor to capture high-level semantics. The extracted features are fed into a per-domain predictive model implemented as a three-layer MLP with hidden dimensions $[128, 64]$ and an output dimension of 10. The encoder and decoder are four-layer MLPs with dimensions $[1024, 512, 128, 32]$. The neural transport operator is implemented as a 128-dimensional linear field network with 5 basis matrices. The learning rate is set to $1 \times 10^{-3}$.

4. **Arxiv** Each paper title is first embedded using a SentenceTransformer encoder, resulting in a 384-dimensional representation. The per-domain predictive model is a three-layer MLP with hidden dimensions $[50, 50]$ and output dimension 10. The encoder and decoder are four-layer MLPs with layer dimensions $[1024, 512, 128, 32]$. The neural transport operator is implemented as a 32-dimensional linear field network with 5 basis matrices. The learning rate is set to $1 \times 10^{-3}$ for all components.

5. **YearBook** The shared feature extractor is a convolutional backbone composed of three convolutional layers with channels $[32, 32, 64]$, each followed by a ReLU activation and a max pooling layer. The output is flattened and passed through a dropout layer. The per-domain predictive model is a three-layer MLP with hidden dimensions $[128, 32]$ and output dimension 2. The encoder and decoder are four-layer MLPs with dimensions $[1024, 512, 128, 32]$. The neural transport operator is implemented as a 32-dimensional linear field network with 5 basis matrices. The learning rate is set to $1 \times 10^{-3}$.

6. **Traffic** The predictive model is a three-layer MLP that takes as input a flattened 96-dimensional vector representing 48 historical inflow and outflow pairs, and outputs a 6-dimensional vector corresponding to 3-step future predictions. The hidden dimension is set

---

[4]https://github.com/Zekun-Cai/NeuralLio

to 64. The encoder and decoder are four-layer MLPs with dimensions $[1024, 512, 128, 32]$. The neural transport operator is implemented as a 32-dimensional linear field network with 5 basis matrices. The learning rate is set to $1 \times 10^{-3}$ for all components.

## A.4 Complexity Analysis

In our framework, all domains share a common feature extractor, while each domain is associated with a domain-specific parameter vector $\theta \in \mathbb{R}^D$. This vector is first encoded into a low-dimensional latent embedding $e$ through a neural autoencoder implemented as an MLP. The embedding $e$ is then updated by a neural transport operator defined over the descriptor space and subsequently decoded back into $\theta$ to produce the generalized model.

The overall complexity of this transformation is $\mathcal{O}(2(Dn + E) + F)$, where $n$ is the width of the first encoder layer, $E$ is the total number of parameters in the remaining layers of the encoder, and $F$ denotes the parameters in the transport and gating modules. Specifically, $F$ includes the field network, which maps the source descriptor to a small set of basis matrices in the latent space, and the coefficient network, which maps descriptor differences to scalar weights over those basis matrices. Since both networks operate in a low-dimensional space (i.e., the same dimension as $e$), their parameter sizes remain small. The gating module is implemented as a lightweight mask vector over descriptor dimensions and adds negligible overhead. In practice, $D$ is also small because most model parameters, such as those in convolutional or transformer-based feature extractors, are shared across domains. Domain-specific parameters are usually applied only to a few final layers. Consequently, the overhead introduced by our framework remains controlled.

## A.5 Ablation Study

To understand the contribution of each core component in our framework, we conduct a series of ablation studies on the 2-Moons and Traffic datasets. We focus on the importance of structure-aware transport design, gating mechanisms, and local chart-based inference. All variants are trained and evaluated under the same protocol as the main experiments.

We achieve the following variants, results can be found in Table 3. (1) **Plain**: replaces the neural Lie transport operator with a plain MLP that directly maps $(z_i, z_j, e_i)$ to $e_j$, termed as $\mathcal{T} - Plain$. The performance of the $\mathcal{T} - Plain$ variant demonstrates the limitations of removing geometric and algebraic constraints, as it consistently underperforms our full model across all datasets. The lack of structural bias causes the transport function to overfit to training tuples,

Table 3: Ablation test results for different datasets.

| Ablation | 2-Moons | Traffic |
|---|---|---|
| $\mathcal{T} - Plain$ | $13.3 \pm 0.4$ | $16.5 \pm 1.0$ |
| $\mathcal{T} - noLie$ | $34.8 \pm 0.5$ | $16.4 \pm 0.3$ |
| w/o Gating | - | $16.3 \pm 0.4$ |
| w/o Chart | - | $16.1 \pm 0.3$ |
| NeuralLio | $\mathbf{3.2 \pm 1.2}$ | $\mathbf{15.1 \pm 0.1}$ |

failing to extrapolate to unseen domains. (2) **noLie**: bypass the exponential mapping that projects Lie algebra elements to the Lie Group. Instead, the descriptor difference $\Delta z$ is first applied to the field network to produce a raw linear operator, which is then applied directly to the source embedding $e_i$ to obtain $e_j$. $\mathcal{T} - noLie$ variant reduces the transformation to a descriptor-conditioned linear mapping without enforcing any group structure. It leads to severe performance degradation, especially on the 2-Moons dataset, and remains suboptimal on the real-world dataset. This highlights the importance of algebraic structure, without which the learned transport fails to align reliably with the underlying domain variation. We further ablate two structural modules designed to handle imperfect descriptors. Disabling the gating mechanism leads to performance drops under noisy or redundant descriptors, indicating its role in suppressing irrelevant dimensions. Removing charting similarly impairs generalization under incomplete descriptors, as transport lacks global consistency. Both components are critical for robust modeling under imperfect descriptor conditions. More detailed evaluations of these modules are presented in Section 5.3.

## A.6 Scalability Analysis

We evaluate the scalability of our framework with respect to two key factors: the number of prediction model parameters and the number of domains. Computation time is measured under increasing model and domain complexity, normalized by the shortest time to provide a consistent basis.

Fig. 7 (a) reports computation time on the fMoW dataset as a function of prediction model size, varied from 80K to 9M parameters by adjusting network depth and width. The growth in runtime is smooth and nearly linear, which aligns with the theoretical complexity described in Section A.4.

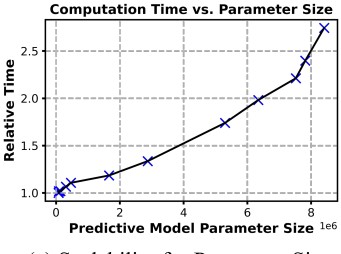

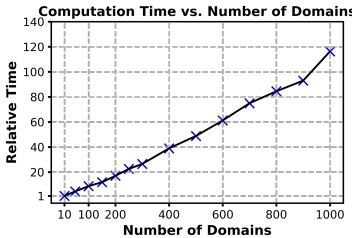

| (a) Scalability for Parameter Size | (b) Scalability for Domain Number |

Figure 7: Scalability analysis w.r.t the number of parameters and the number of domains.

Fig. 7 (b) presents the runtime as a function of the number of domains, evaluated on the 2-Moons dataset with domain counts ranging from 10 to 1000. Domains are sampled from the descriptor space with a fixed number of data points per domain to ensure comparability. The runtime increases linearly with domain count, reflecting the scalability of our per-domain inference pipeline and supporting its applicability to large-scale domain generalization tasks.

Table 4: Cost of training and testing time.

| Model | Train Time (s) | Test Time (s) |
|---|---|---|
| CIDA | 711 | 0.6 |
| DRAIN | 144 | 0.2 |
| TKNets | 1250 | 0.3 |
| Koodos | 540 | 0.1 |
| Ours | 694 | 0.1 |

We further benchmark the wall-clock training time of our method against representative baselines that explicitly model inter-domain relationships, including CIDA [51], DRAIN[2], TKNets [59], and Koodos [6]. To support a broader set of baseline methods, we conduct this comparison on the YearBook dataset. All models are trained under identical hardware and epoch configurations. As shown in Table 4, our method matches the training efficiency of prior structure-aware models (e.g., Koodos, CIDA) while achieving the lowest testing cost. This reflects a well-balanced trade-off of our model between computational efficiency and effectiveness.

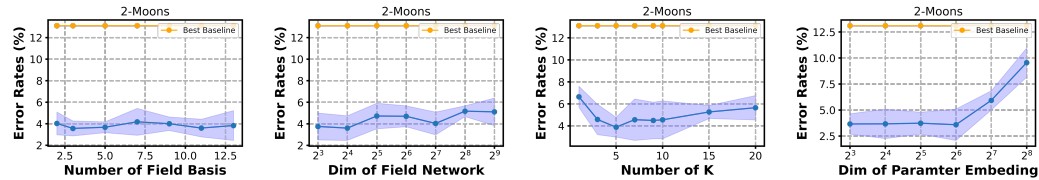

Figure 8: Sensitivity analysis. From left to right: number of field basis vectors used in the transport operator, hidden dimension of the field network, number of neighbors $K$ in the local chart module, and the dimension of the parameter embedding $e$.

## A.7 Sensitivity Analysis

We conduct a sensitivity analysis on the 2-Moons dataset to evaluate the robustness of our method to key hyperparameters. Specifically, we vary four components: (1) the number of field network $f_v^k$ in the operator, denoted as the *Number of Field Basis*; (2) the hidden dimension of the MLP implementing the field network, denoted as the *Dim of Field Network*; (3) the number of neighbors $K$ used in the local chart module; and (4) the embedding dimension of predictive model parameters $e$ after encoding, denoted as the *Dim of Parameter Embedding*.

As shown in Fig. 8, our model exhibits strong robustness across a broad range of values. The error rate remains stable when varying the number of basis vectors and the hidden dimension of the field network, indicating flexibility in the transport capacity. For $K$, performance is consistent for moderate neighborhood sizes (e.g., $K = 5 \sim 10$), while very small $K$ leads to degradation due to

under-smoothing. For the embedding dimension of $e$, we observe an optimal range below 128, which show that overly large values introduce overparameterization. These results show that our model outperforms the baseline model over a wide range of hyperparameters, confirming the robustness and adaptability of our framework.

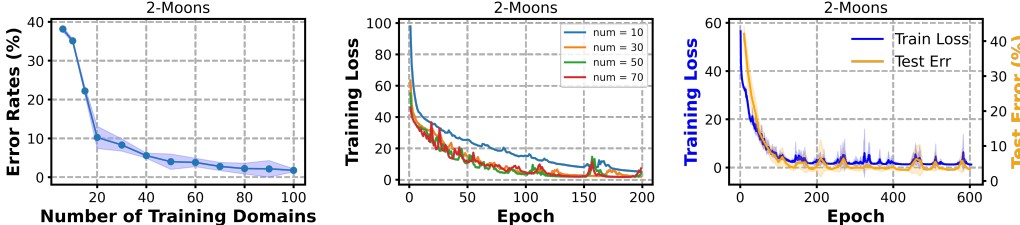

Figure 9: Convergence analysis. **Left:** Test error (%) on 2-Moons as a function of the number of training domains. **Middle:** Training loss curves for different numbers of training domains. **Right:** Training loss and test error over long epochs.

## A.8   Convergence Analysis

To evaluate the learning behavior and convergence properties of our framework, we conduct a series of controlled experiments on the 2-Moons dataset. We design three complementary analyses: (1) convergence with respect to the number of training domains, (2) convergence trajectories across epochs under varying the number of domains, and (3) joint dynamics of training loss and test performance over extended training.

**Convergence results across different domain counts.**   We evaluate convergence under increasing numbers of training domains on the 2-Moons dataset. Specifically, we vary the number of training domains from 5 to 100, while fixing the test domains on a predefined mesh grid to ensure consistent evaluation. As shown in the left panel of Fig. 9, the test error drops rapidly as the number of training domains increases. With as few as 20 training domains, the model already achieves a test error below 10%, and beyond 50 domains, the error consistently falls below 5%. At 80 domains, the model shows near-perfect generalization. These results demonstrate that our model can recover the underlying parameter field even under very sparse supervision, and increasing the number of training domains is crucial for capturing the variation structure.

**Convergence trajectory across different domain counts.**   The center panel of Fig. 9 illustrates the training loss trajectories over epochs under different numbers of training domains (10,30,50,70). Across all settings, the loss consistently decreases and stabilizes after around 120 epochs, demonstrating the convergence stability of our framework. Moreover, as the number of domains increases, the model not only converges to a lower final loss (insight from the previous paragraph) but also reaches convergence more rapidly. This confirms that greater domain coverage not only improves model accuracy but also accelerates optimization.

**Joint dynamics of training loss and test performance.**   The right panel of Fig. 9 plots both the training loss and test error under our main experimental setting, with over 600 epochs. The model converges rapidly and maintains consistent test performance throughout training, showing no signs of performance degradation. This stability arises from the structural constraints imposed by the architecture, which regularize domain-wise variation and prevent the model from memorizing individual domains. This observation aligns with our theoretical analysis in Property 2, suggesting that enforcing structural constraints naturally limits model overfitting and supports robust generalization.

## A.9   Limitations

Our framework assumes the availability of continuous descriptors to model domain variation. Two practical challenges arise in this context. First, many domain generalization tasks are still formulated under discrete domain assumptions. However, we contend that such seemingly discrete observations reflect limited observations over an implicit continuous process. For example, real, cartoon, and sketch domains likely lie along a visual continuum from realism to abstraction; formal and informal text styles mark the endpoints of a stylistic spectrum; and related languages such as English, German,

and Dutch reflect gradual linguistic evolution. The success of recent diffusion models also supports this view. For such cases, the challenge is not discontinuity, but how to coordinate observed domains within the implicit continuous space. Second, in more extreme scenarios, descriptors may be entirely unavailable or unobservable. While this is uncommon in structured environments, automatic descriptor inference is possible. Recent advances in contrastive representation learning and adversarial domain embedding provide promising tools to estimate latent descriptors directly from the input. These mechanisms can be readily integrated into our framework as additional descriptor discriminators or encoder networks.

Overall, we view domain descriptor construction as a key and promising direction. Building on our current framework, such extensions can further expand the applicability of continuous domain generalization to more weakly-supervised environments.

# B  Optimization Details

The joint training strategy follows the general optimization paradigm established in [6], which has been shown to be principled and empirically validated for coordinating the temporal evolution of model parameters. Our method generalizes its idea to a structural transport operator across descriptor space, extending its applicability beyond purely temporal settings. Specifically, we jointly optimize the per-domain parameters $\theta(z_i)$, encoder $f_e$, decoder $f_d$, neural transport operator $\mathcal{T}$, and gating mechanism. For each training descriptor $z_i$, we supervise its neighborhood $\mathcal{N}(z_i)$ by aligning predictions, latent representations, and transported parameters between $(z_i, z_j)$ pairs. This formulation ensures local consistency and smooth transitions across the parameter manifold. During inference, we identify the nearest training descriptor for a test domain and apply the transport operator to infer its model parameters. The detailed training and inference procedure are summarized in Algorithm 1.

---

**Algorithm 1:** Training and Inference Procedure of NeuralLio

---

**Input:** Domain set $\{(X_i, Y_i)\}_{i=1}^{N}$ with descriptors $\{z_i\}_{i=1}^{N}$; Predictive model architecture $g$;
     Test data $X_s$ with descriptor $z_s$
**Output:** Learned parameters $\{\theta(z_i)\}_{i=1}^{N}$; Encoder $f_e$, Decoder $f_d$, Operator $\mathcal{T}$; Gating **g**;
     Test prediction $\hat{Y}_s$

                                                    // -- Training phase --
**Initialize modules:** initialize per-domain parameters $\{\theta(z_i)\}_{i=1}^{N}$, encoder $f_e$, decoder $f_d$,
 transport operator $\mathcal{T}$, and gating **g**
**Build neighborhoods:** for each $z_i$, compute $k$-nearest neighbors $\mathcal{N}(z_i)$ using Euclidean distance
**foreach** *training iteration* **do**
 
    Sample minibatch $\mathcal{B}$ of domain indices
    **foreach** $i \in \mathcal{B}$ **do**
 
       Fetch $(X_i, Y_i, \theta(z_i))$; encode $e(z_i) = f_e(\theta(z_i))$; reconstruct $\hat{\theta}(z_i) = f_d(e(z_i))$
       Compute losses: $\mathcal{L}_{\text{pred}}^i = \ell(g(X_i; \theta(z_i)), Y_i), \quad \mathcal{L}_{\text{recon}}^i = \|\hat{\theta}(z_i) - \theta(z_i)\|^2$
       **foreach** $z_j \in \mathcal{N}(z_i)$ **do**
 
          Fetch $(X_j, Y_j, \theta(z_j))$
          Gating: $\mathbf{m}(z_i) = \text{Sigmoid}(Wz_i) \odot \mathbf{w}$, $\tilde{z}_i = z_i \odot \mathbf{m}(z_i)$, $\tilde{z}_j = z_j \odot \mathbf{m}(z_i)$
          Predict $\hat{e}(z_j) = \mathcal{T}(e(z_i), \tilde{z}_i, \tilde{z}_j)$; decode $\hat{\theta}(z_j) = f_d(\hat{e}(z_j))$
          Compute $\mathcal{L}_{\text{pred}}^{ij} = \ell(g(X_j; \hat{\theta}(z_j)), Y_j)$
          Compute $\mathcal{L}_{\text{consist}}^{ij} = \|\hat{\theta}(z_j) - \theta(z_j)\|^2$
          Compute $\mathcal{L}_{\text{embed}}^{ij} = \|\hat{e}(z_j) - f_e(\theta(z_j))\|^2$
 
    Update all modules with total loss:
    $\mathcal{L} = \sum_i (\mathcal{L}_{\text{pred}}^i + \mathcal{L}_{\text{recon}}^i) + \sum_{(i,j)} (\mathcal{L}_{\text{pred}}^{ij} + \mathcal{L}_{\text{consist}}^{ij} + \mathcal{L}_{\text{embed}}^{ij})$
 
                                                    // -- Inference phase --
Find nearest $z_i$ to $z_s$ and fetch $\theta(z_i)$; encode $e(z_i) = f_e(\theta(z_i))$
Gating: $\mathbf{m}(z_i) = \text{Sigmoid}(Wz_i) \odot \mathbf{w}$, $\tilde{z}_s = z_s \odot \mathbf{m}(z_i)$
Generalize: $\hat{e}(z_s) = \mathcal{T}(e(z_i), \tilde{z}_i, \tilde{z}_s)$; $\hat{\theta}(z_s) = f_d(\hat{e}(z_s))$
Predict: $\hat{Y}_s = g(X_s; \hat{\theta}(z_s))$

---

