# OpenReview forum: "Continuous Domain Generalization"
_NeurIPS.cc/2025/Conference — NeurIPS 2025 poster_

### Official Review · Reviewer_DmZF · 2025-06-23

**Clarity:** 4
**Significance:** 3
**Originality:** 3
**Rating:** 4
**Confidence:** 2

**Summary:**

This paper extends predictive modeling frameworks by introducing a generalization from a single latent variable (e.g., time) to multiple latent variables. To this end, the authors propose a neural Lie transport operator designed to ensure smooth transitions over a learned manifold, subject to certain regularity conditions. To handle imperfect input descriptions, such as those that are noisy or sparse, the method incorporates gating mechanisms and local coordinate charts to maintain robustness. Experimental results on both synthetic and real-world datasets demonstrate the effectiveness and generalizability of the proposed approach.

**Questions:**

Questions:

1. In case of the Yearbook dataset, if the descriptor is a single-axis, which existing baselines assume, why does the proposed method perform the best even in these scenarios?
2. It would be helpful for the authors to provide a more detailed, intrinsic interpretation of the visualizations presented in Figures 2 and 3. Beyond evaluating the quality of the decision boundary, is there any additional insight that the visualization in Figures 2 and 3 is intended to convey? For example, in Figure 3, how does the visualization by altering the direction of the moon contribute to understanding the learned manifold or the behavior of the proposed model?
3. Does the performance enhancement difference, such as high performance gain in the synthetic dataset, while relatively low gain in the real-world dataset, relate to non-degenerate representation of the descriptions?
4. In equation 6, how can $(z_j^k - z_i^k) \in \mathbb{R}^d$ and $f_v^k(z_i) \in \mathbb{R}^{m \times m}$ be multiplied? Could the authors clarify about this equation?

**Ethical Concerns:**

["NO or VERY MINOR ethics concerns only"]

**Final Justification:**

Most of my concerns have been adequately addressed in the rebuttal, and I intend to maintain my positive evaluation.

**Limitations:**

yes

**Paper Formatting Concerns:**

.

**Quality:**

3

**Strengths And Weaknesses:**

Strengths:

- The paper is easy to follow.
- The motivation for extending predictive models to incorporate multiple latent variables, rather than relying solely on time as a single axis of variation, is well-justified and compelling.
- The performance of the synthetic dataset affects the need to incorporate multiple latent variables.

Weaknesses:

- Some visualizations are challenging to interpret (Figure 2).
- The performance gain on the real-world dataset is marginal compared to the synthetic dataset.
- Assuming non-degeneracy may be unrealistic in real-world settings, where input descriptors are frequently noisy or ambiguous.

---

> ### Author Rebuttal · Authors · 2025-07-30
>
> Thanks for reviewing and acknowledging our work. Please find our response as follows:
> > W1 & Q2. Some visualizations are challenging to interpret (Figure 2). It would be helpful for the authors to provide a more detailed, intrinsic interpretation of the visualizations presented in Figures 2 and 3. Beyond evaluating the quality of the decision boundary, is there any additional insight that the visualization in Figures 2 and 3 is intended to convey? For example, in Figure 3, how does the visualization by altering the direction of the moon contribute to understanding the learned manifold or the behavior of the proposed model?
> 1) Figures 2 and 3 are not only intended to show decision boundary quality, but more importantly, to provide intrinsic insights into the learned manifold and **how model parameters co-evolves with domain distribution variation**:
> 2) Figure 2 left provides a visualization of the domain descriptor space of the 2-Moons dataset, where the two axes represent rotation (periodicity) and scaling (monotonicity). The background colors represent the periodicity and monotonicity. The black dots indicate training domains, while the white insets show that the black dots correspond to domain data. We randomly select four test domains (marked with blue crosses) to show the prediction of the baseline methods in Figure 2, right. All baseline models generalize very poorly.
> 3) In Figure 3, the left plot visualizes the learned **global parameter manifold** via PCA, showing a ‘twisted sheet-like manifold surface’ that aligns with the periodicity of rotation and the monotonicity of scaling. **The right plot shows that varying the domain descriptor corresponds to traversing on the manifold surface**. The continuous evolution of decision boundaries along single and mixed directions confirms that the learned manifold is **continuous and geometrically consistent**. In particular, the mixed-direction case highlights the model’s ability to **capture multiple types of variations jointly**.
> 4) We will include this clarification in the camera-ready version.
>
> > W2 & Q3. The performance gain on the real-world dataset is marginal compared to the synthetic dataset. Does the performance enhancement difference, such as high performance gain in the synthetic dataset, while relatively low gain in the real-world dataset, relate to non-degenerate representation of the descriptions?
> 1) While the performance gains are more pronounced on synthetic datasets, this does not diminish the significance of our gains on real-world benchmarks: our method achieves 12% on average over descriptor-agnostic baselines, and 9% on average over SOTA approaches. Such gains are widely regarded as significant in the domain generalization literature.
> 2) Synthetic datasets are intentionally designed with clear and interpretable domain variations to rigorously evaluate a model’s mathematical behavior. Our method nearly solves these benchmarks, demonstrating strong theoretical soundness. However, directly comparing gains on synthetic and real-world datasets is unfair. While synthetic benchmarks isolate specific mathematical challenges, real-world datasets involve a coupling of high-dimensional, noisy, and entangled factors, which makes improvements inherently more difficult.
> 3) Naturally, descriptor quality affects performance. Our method is explicitly designed to mitigate such imperfections (Section 4.3). As shown in Section 5.3, improving descriptor quality can further enhance model performance. We regard this as an important follow-up direction built upon the current work.
>
> > W3. Assuming non-degeneracy may be unrealistic in real-world settings, where input descriptors are frequently noisy or ambiguous.
> 1) We appreciate the comment. To clarify, **Theorem 1 does not assume the descriptors used in practice**. Rather, it expresses that a non-degenerate descriptor space theoretically exists and characterizes the model parameter structure. Such a theoretical descriptor space is grounded in reality, as distribution shifts are ultimately governed by a finite set of latent factors.
> 2) Practical descriptors are imperfect. We fully acknowledge this, but **practical descriptors serve as projections of the theoretical space**, so having partial descriptors is still better than having none. Moreover, the underlying manifold established in Section 4.1 **exists independently of practical descriptor quality**. Our method explicitly addresses the imperfect problem in Section 4.3. Experiments in real-world datasets and Section 5.3 further demonstrate that these partial descriptors are informative and effective.
> 3) We will revise the statement of Theorem 1 to clarify this point and remove ambiguity.
>
> > Q1. In case of the Yearbook dataset, if the descriptor is a single-axis, which existing baselines assume, why does the proposed method perform the best even in these scenarios?
>
> Our method is **the only one to jointly encode both geometric continuity and algebraic consistency**, leading to a more complete modeling of the evolution structure, no matter in single-axis or multi-axis scenarios:
> 1) Traditional descriptor-agnostic models treat domains as independent entities, ignoring any relational structure.
> 2) CIDA utilizes distances between domain indices to model domain relationships; however, distance functions induce only a metric space, and they cannot capture the smoothness, directional derivatives, or curvature inherent in geometry.
> 3) TKNets and DRAIN model temporal dependencies via sequence-based mechanisms, but lack both geometric and algebraic constraints.
> 4) Koodos recognizes the continuity of distributional shifts, yet it does not incorporate algebraic structure into parameter evolution.
>
> > Q4. In equation 6, how can they be multiplied?
>
> We confirm that Equation (6) is mathematically correct. Specifically, although $(z_j - z_i) \in \mathbb{R}^d $, the term $(z_j^k - z_i^k)$ is a scalar (i.e., the $k$-th coordinate difference of the $d$-dimensional vector), and $f_v^k(z_i) \in \mathbb{R}^{m \times m}$, so their product is a valid scalar-matrix multiplication.

---

> > ### Comment · Reviewer_DmZF · 2025-08-02
> >
> > Thank the authors for the rebuttal.
> >
> > Most of my questions are well addressed, and I have one more question: whether the proposed method also satisfies Identity, Associativity, invertibility, or other algebraic properties as in Definition 2 of the main paper.

---

> > > ### Author Response · Authors · 2025-08-03
> > >
> > > Thank you for your thoughtful follow-up.
> > >
> > > Yes, we have investigated the algebraic properties of our learned transport operator. Specifically:
> > >
> > > 1) NeuralLTO is grounded in Lie group theory and is defined via exponential maps in Lie algebras, which **theoretically satisfy group properties by definition**.
> > >
> > > 2) To validate this empirically, we obtain the model parameters $\theta(z)$ for all test domains on 2-Moons, and conduct the following evaluations:
> > >
> > > - **Identity**: For each test descriptor $z_i$, we apply $\theta(z_i) \xrightarrow{z_i \rightarrow z_i} \theta'(z_i)$ and evaluate the cosine similarity: $\cos(\theta(z_i), \theta'(z_i))$.
> > >
> > > - **Associativity**: We consider all triplet combinations $(z_i, z_j, z_k)$ from the test descriptors. For each triplet, we get two results: $\theta(z_i) \xrightarrow{z_i \rightarrow z_j} \theta'(z_j) \xrightarrow{z_j \rightarrow z_k} \theta'(z_k)$ and $\theta(z_i) \xrightarrow{z_i \rightarrow z_k} \theta''(z_k)$, then compute $\cos(\theta'(z_k), \theta''(z_k))$.
> > >
> > > - **Invertibility**: We consider all pairs $(z_i, z_j)$ from the test descriptors. For each pair, we apply $\theta(z_i) \xrightarrow{z_i \rightarrow z_j} \theta'(z_j) \xrightarrow{z_j \rightarrow z_i} \theta'(z_i)$, then compute $\cos(\theta(z_i), \theta'(z_i))$.
> > >
> > > - **Closure** is inherently satisfied, as the transport operator always maps within the valid parameter space.
> > >
> > >
> > > **Results**:
> > >
> > > | Property       | Cosine Similarity |
> > > |----------------|-------------------|
> > > | Identity       | 99.9%             |
> > > | Associativity  | 99.1%             |
> > > | Invertibility  | 98.3%             |
> > >
> > > These results indicate that the learned NeuralLTO operator respects the group axioms with high fidelity, thus empirically validating the theoretical formulation.
> > >
> > > We appreciate your constructive comments and will incorporate these results into the camera-ready version.

---

> > > > ### Comment · Reviewer_DmZF · 2025-08-04
> > > >
> > > > I appreciate the authors' detailed response. Most of my concerns and questions have been addressed. I will continue the discussion with the other reviewers as needed.

---

> > > > > ### Author Response · Authors · 2025-08-04
> > > > >
> > > > > Thank you for your positive feedback and for acknowledging our work. We appreciate your decision and are glad that our responses helped address your concerns.

---

### Official Review · Reviewer_sRq5 · 2025-06-30

**Clarity:** 2
**Significance:** 2
**Originality:** 3
**Rating:** 4
**Confidence:** 4

**Summary:**

The paper introduces Continuous Domain Generalization (CDG), which assumes each domain is specified by a real-valued descriptor vector (e.g., climate, season, socioeconomic indices) rather than by a discrete label. To address the issue of time-evolving pattern in each domain, the authors proposed a Neural Lie Transport Operator to capture the complex, multi-dimensional nature of real-world variation

**Questions:**

- Why the existing methods only capture partial domain relationships, and they are limited in scope and fail to capture the full structure of continuous domain variation?
- How does the model perform on larger datasets, such as domain net, which is commonly used by other DG methods.
- How does the proposed method compare to continual learning? And why the setting of DG is needed.
- What is the computation time if the proposed method is applied on a larger network (e.g. ResNet)?
- The motivation for using Structured Constrained Domain Transport rather than PDE is not clear.

**Ethical Concerns:**

["NO or VERY MINOR ethics concerns only"]

**Final Justification:**

Most of my concerns has been addressed in the rebuttal

**Limitations:**

yes

**Quality:**

3

**Strengths And Weaknesses:**

- Strength
  - Solid theoretical background
  - Covers divers application domain
- Weakness
  - Benchmark dataset and backbone network is small.
  - Motivation is not quite clear for the proposed adaptation scenario (continual DG) and the proposed transport operator
  - Relationship to continual learning is not very clear.

---

> ### Author Rebuttal · Authors · 2025-07-30
>
> Thanks for reviewing. Please find our response as follows:
> > W1. Benchmark dataset and backbone network is small.
>
> **Dataset**: Our evaluation includes four large-scale real-world datasets with uncontrolled and multi-factor domain shifts. All datasets are widely recognized for distribution shift benchmark: fMoW, ArXiv, and Yearbook are from paper in NeurIPS21 DB Track [55] (citation #121), one of the most comprehensive and widely adopted evaluations for distribution shift. Traffic from paper in KDD19 [41](citation #647), and has been used in numerous urban analytics and spatiotemporal generalization studies.
>
> **Backbone**: To address concerns regarding the backbone size, we further evaluate our method on the fMoW dataset using ResNet-50 pretrained on ImageNet as the backbone. As shown in the table below, our method continues to outperform all descriptor-agnostic and descriptor-aware baselines, validating that our framework can generalize well under more complex network configurations.
>
> |Type | Method     | fMoW (ResNet-50 as backbone) |
> |-----------------|------------|-------------------|
> | Descriptor-Agnostic | ERM        | 27.7 ± 1.6        |
> |                 | IRM        | 41.5 ± 2.8        |
> |                 | V-REx      | 32.1 ± 3.6        |
> |                 | GroupDRO   | 28.6 ± 1.9        |
> |                 | Mixup      | 27.1 ± 1.5        |
> |                 | DANN       | 26.0 ± 0.7        |
> |                 | MLDG       | 29.2 ± 1.0        |
> |                 | CDANN      | 27.6 ± 0.9        |
> |                 | URM        | 26.9 ± 1.0        |
> | Descriptor-Aware   | ERM-D      | 28.9 ± 1.8        |
> |                 | NDA        | 31.2 ± 1.4        |
> |                 | CIDA       | 27.1 ± 0.9        |
> |                 | **Ours**   | **24.5 ± 0.5**    |
>
>
> > W2. Motivation is not quite clear for the proposed adaptation scenario (continual DG)
>
> 1) We would like to first clarify that our work 'Continuous Domain Generalization' is not 'continual DG'. **'Continuous' and 'Continual' have different meanings in the context of DG**. The term “continuous” refers to a process that changes smoothly, while “continual” describes something that occurs repeatedly or intermittently over time.
> 2) The motivation for CDG lies in that domain distribution shifts are **continuous over the underlying governed factor space** (e.g., space, time, environment), yet existing methods largely treat domains as discrete entities. Addressing this gap is both principled and practical:
> - **Principled motivation**: CDG is grounded in the **continuum hypothesis**, widely adopted in physics and engineering: every real-world macroscopic system evolves smoothly, with its quantities modeled as continuous fields governed by differential equations.
> - **Practical motivation**: In the third paragraph of Introduction, we decompose a real-world example, satellite land-use classification to illustrate how gradual environmental and developmental changes induce smooth shifts in distribution. We further present three additional realistic cases (i.e., healthcare, urban analytics, and visual recognition) to support the broad applicability of CDG.
> 3) Prior DG methods ignore this continuum and treat domains as independent entities, leading to fragmented and inefficient generalization. These limitations strongly motivate our proposed formulation.
>
> We appreciate your constructive comments and will incorporate the above content into the camera-ready version to improve clarity.
>
> > W2 & Q5. Motivation is not quite clear for the proposed transport operator. The motivation for using Structured Constrained Domain Transport rather than PDE is not clear.
>
> We clarify that although model parameters $\theta$ follow a PDE-like evolution as Eq. 2, CDG poses the following challenges:
> - (c1) The ground-truth $\theta_i$ for each training domain is unobservable. They are implicit and can only be obtained through iterative optimization.
> - (c2) The training domains are sparsely and non-uniformly sampled in the descriptor space.
> - (c3) The analytical form of the distribution shift governing PDE is unknown.
>
> As a result, existing standard PDE-solving techniques are not applicable:
> - Numerical methods require known boundary or initial values (unavailable due to c1);
> - Separation of variables methods rely on decomposing the PDE into a set of ODEs along a continuous direction (infeasible due to c2);
> - Physics-Informed Neural Networks require explicit PDE forms to define the loss (unavailable due to c3).
>
> These limitations motivate our use of neural operators. Directly learn mappings between function spaces, which are widely studied in scientific machine learning.
>
> We have discussed this rationale in the first paragraph of Section 4.2, and will further clarify it in the camera-ready version.
>
> > W3 & Q3. Relationship to continual learning is not very clear. How does the proposed method compare to continual learning? And why the setting of DG is needed.
>
> 1) We respectfully clarify that continual learning and continuous domain generalization refer to fundamentally different problems:
> - **Continual learning** assumes a temporal task sequence, where the objective is how to learn on the current domain while retaining performance on previously learned ones, typically addressing the catastrophic forgetting problem under constraints such as no access to past data.
> - **Continuous domain generalization** addresses scenarios where domains vary continuously over a descriptor space, and the objective is to model how predictive models co-evolve as a function of distribution variation, enabling generalization to any domain in the continuum.
> 2) Although both involve the word ’continue’, they differ substantially in formulation, assumptions, and objectives.
>
> > Q1. Why the existing methods only capture partial domain relationships, and they are limited in scope and fail to capture the full structure of continuous domain variation?
>
> 1) As we discussed earlier, real-world processes are continuous, resulting domain distribution shifts are continuous. This implies the existence of a structural mapping: how predictive models $\theta$ continuously evolve as a function of the distribution variation. **Capturing this globally evolving functional relationship** defines what we refer to as the full structure of continuous domain variation.
> 2) By contrast, prior methods are limited in scope:
> - Traditional DG methods ignore all domain relations but assume domains are categorical and independent.
> - The temporal DG method compresses the general multi-dimensional variation space into a one-dimensional temporal sequence. This reduction prevents it from capturing the intrinsic geometry of domain shifts.
> - Some methods leverage partial domain index relationships.  AdaGraph [37] constructs graphs over domain indices; however, graphs are discrete and non-differentiable, making them incapable of modeling a global continuous mapping over the domain space. CIDA [51] utilizes domain indices distances to model domain relationships; however, distance functions induce only a metric space, and they cannot capture the smoothness, directional derivatives, or curvature inherent in geometry.
> 3) Our method, by modeling $\theta$ as a continuous function learned via a structurally constrained neural operator, enables capturing the **full geometric structure** of domain variation beyond the limitations of discrete or partial relationship methods.
>
> > Q2. How does the model perform on larger datasets, such as domain net, which is commonly used by other DG methods.
>
> While DomainNet is a popular benchmark in traditional DG, it is not suitable for our problem setting due to the absence of domain descriptors. We acknowledge that handling distribution shifts without explicit descriptors is an important and practical extension. We view this as a natural future direction of our work, and have discussed it in detail in the Limitation section.
>
> > Q4. What is the computation time if the proposed method is applied on a larger network (e.g. ResNet)?
>
> We report runtime results on fMoW using ResNet-50 (pretrained on ImageNet). Since baselines such as DRAIN, TK-Nets, and Koodos reported in Table 3 (running time analysis) are not compatible with fMoW, CIDA remains the only applicable descriptor-aware baseline.
>
> | Method | Train Time (min) | Test Time (s) |
> |--------|------------------|----------------|
> | ERM    | 17               | 6.0            |
> | CIDA   | 176              | 12.0           |
> | Ours   | 128              | 8.1            |
>
> These results indicate that our method maintains competitive efficiency compared to existing domain relationship modeling approaches.

---

> > ### Comment · Reviewer_sRq5 · 2025-08-04
> >
> > Thank authors for the detailed rebuttle, most of my concerns have been addressed.

---

> > > ### Author Response · Authors · 2025-08-04
> > >
> > > Thank you for your thoughtful consideration and for recognizing our efforts. Your feedback has been invaluable in refining our work.

---

### Official Review · Reviewer_aVTi · 2025-06-30

**Clarity:** 2
**Significance:** 1
**Originality:** 3
**Rating:** 1
**Confidence:** 4

**Summary:**

Authors propose a new task named Continuous Domain Generalization (CDG) and a method named Neural Lie Transport Operator (NeuralLTO) to address the new task. They conduct theoretical justifications and empirical analyses.

**Questions:**

See weaknesses.

**Ethical Concerns:**

["NO or VERY MINOR ethics concerns only"]

**Final Justification:**

I appreciate efforts of authors made in rebuttal. However, the paper still seems to be far below the bar of the conference to me.

For theoretical analyses, even if this is not an assumption of practical descriptors, such an assumption still makes the proof of theorem too straightforward, which significantly weakens the theoretical contributions.

For experimental improvement, it is too tricky to look at relative improvement mentioned in the rebuttal.

For model backbones, I suggest all experiments should be conducted again with modern backbones like ResNet-50 or ViTs. Otherwise, such experimental results are not really persuasive nowadays.

Therefore, I decide to keep my score.

**Limitations:**

See weaknesses.

**Quality:**

1

**Strengths And Weaknesses:**

Strengths: The problem is overall new, with some theoretical and empirical analyses as support.

Weaknesses:

- The theoretical analyses seem to be insufficient. Theorem 1 assumes that the d-dimensional descriptor provides a complete and non-degenerate representation, which makes the conclusion and proof a little trivial. Meanwhile, it could be a strong assumption that a descriptor providing a complete and non-degenerate representation is available in reality.
- Experiments do not seem quite convincing.
  - Improvements over baselines on most real-world datasets (FMoW, ArXiv, Traffic) are marginal, with less than 1 pp.
  - More advanced descriptor-agnostic DG methods (i.e. standard DG methods) could be added as baselines for comparison.  For example, Fishr [1] and SWAD [2].
  - Network backbones used in experiments (listed in Appendix A.3) are too small, making the experiments toy. In standard DG literature, ResNet-50 and ViT are commonly adopted, while only a three-layer CNN is adopted in experiments of image classification (MNIST, fMoW, YearBook). It is reasonable to use small models like MLP for tabular data like 2-Moon and Traffic since larger models usually do not work well on tabular data. However, for image tasks, using such small models does not really make sense.
- The problem of CDG assumes access to multi-dimensional descriptors. However, such metadata of domains is often not available in real-world applications. This could severely limit the application range of the proposed problem setting and method, even making the setting a bit artificial. More realistic examples could be provided to justify the practical usage of the proposed problem setting and method.
- The title of the paper in the openreview system is "Continuous Domian Generalization", which is an eye-catching typo. Such a severe typo weakens the reliability of the paper to some extent.



### References

[1] Rame A, Dancette C, Cord M. Fishr: Invariant gradient variances for out-of-distribution generalization[C]//International Conference on Machine Learning. PMLR, 2022: 18347-18377.

[2] Cha J, Chun S, Lee K, et al. Swad: Domain generalization by seeking flat minima[J]. Advances in Neural Information Processing Systems, 2021, 34: 22405-22418.

---

> ### Author Rebuttal · Authors · 2025-07-30
>
> Thanks for reviewing. Please find our response as follows:
> > W1. The theoretical analyses seem to be insufficient. Theorem 1 assumes that the d-dimensional descriptor provides a complete and non-degenerate representation, which makes the conclusion and proof a little trivial. Meanwhile, it could be a strong assumption that a descriptor providing a complete and non-degenerate representation is available in reality.
> W3. The problem of CDG assumes access to multi-dimensional descriptors. However, such metadata of domains is often not available in real-world applications. This could severely limit the application range of the proposed problem setting and method, even making the setting a bit artificial. More realistic examples could be provided to justify the practical usage of the proposed problem setting and method.
> 1) We would clarify that **Theorem 1 does not assume the descriptors used in practice**. Rather, it expresses that a complete, non-degenerate descriptor space theoretically exists and characterizes the model parameter structure. Such a theoretical descriptor space is grounded in reality, as distribution shifts are ultimately governed by a finite set of latent factors.
> 2) Practical descriptors are imperfect. We fully acknowledge this, but  **practical descriptors serve as projections of the theoretical space**, so having partial descriptors is still better than having none. Moreover, the underlying manifold established in Section 4.1 **exists independently of practical descriptor quality**. Our method explicitly addresses the imperfect problem in Section 4.3. Experiments in real-world datasets and Section 5.3 further demonstrate that these partial descriptors are informative and effective.
> 3) We will revise the statement of Theorem 1 to clarify this point and remove ambiguity.
>
> > W2.1 Improvements over baselines on most real-world datasets (FMoW, ArXiv, Traffic) are marginal, with less than 1 pp.
>
> We respectfully clarify that in terms of relative improvement, our method achieves 12% on average over descriptor-agnostic baselines, and 9% on average over SOTA approaches. Such gains are widely regarded as significant in the domain generalization literature.
>
> > W2.2 More advanced descriptor-agnostic DG methods (i.e. standard DG methods) could be added as baselines for comparison. For example, Fishr [1] and SWAD [2].
>
> We have added the more recent and standard DomainBed benchmark method Fishr to the experimental comparisons across all six benchmark datasets. The updated results are shown below:
>
> | Method | 2-Moons | MNIST | fMoW | ArXiv | Yearbook | Traffic |
> |--------|-------------|-----------|----------|-----------|---------------|--------------|
> | Fishr  | 34.9 ± 0.2 | 29.5 ± 1.4 | 37.5 ± 0.8 | 35.5 ± 0.1 | 7.1 ± 0.9 | 18.7 ± 0.2 |
> | **Ours**   | **3.2 ± 1.2** | **9.5 ± 1.1** | **35.9 ± 0.5** | **34.7 ± 0.4** | **4.8 ± 0.3** | **15.1 ± 0.1** |
>
> These results confirm that our method continues to achieve state-of-the-art performance across all real-world and synthetic datasets.
>
> > W2.3 Network backbones used in experiments (listed in Appendix A.3) are too small, making the experiments toy. In standard DG literature, ResNet-50 and ViT are commonly adopted, while only a three-layer CNN is adopted in experiments of image classification (MNIST, fMoW, YearBook). It is reasonable to use small models like MLP for tabular data like 2-Moon and Traffic since larger models usually do not work well on tabular data. However, for image tasks, using such small models does not really make sense.
>
> To address concerns regarding the backbone size, we further evaluate our method on the fMoW dataset using ResNet-50 pretrained on ImageNet as the backbone. As shown in the table below, our method continues to outperform all descriptor-agnostic and descriptor-aware baselines, validating that our framework can generalize well under more complex network configurations.
>
> |Type | Method     | fMoW (ResNet-50 as backbone) |
> |-----------------|------------|-------------------|
> | Descriptor-Agnostic | ERM        | 27.7 ± 1.6        |
> |                 | IRM        | 41.5 ± 2.8        |
> |                 | V-REx      | 32.1 ± 3.6        |
> |                 | GroupDRO   | 28.6 ± 1.9        |
> |                 | Mixup      | 27.1 ± 1.5        |
> |                 | DANN       | 26.0 ± 0.7        |
> |                 | MLDG       | 29.2 ± 1.0        |
> |                 | CDANN      | 27.6 ± 0.9        |
> |                 | URM        | 26.9 ± 1.0        |
> | Descriptor-Aware   | ERM-D      | 28.9 ± 1.8        |
> |                 | NDA        | 31.2 ± 1.4        |
> |                 | CIDA       | 27.1 ± 0.9        |
> |                 | **Ours**   | **24.5 ± 0.5**    |
>
> > W4. typo
>
> We sincerely apologize for the typo. This error exists only in the OpenReview system and does not appear in the actual manuscript. We will correct it accordingly. We emphasize that this mistake does not affect the technical content, contributions, or reliability of the manuscript.

---

> > ### Comment · Reviewer_aVTi · 2025-08-06
> >
> > I appreciate efforts of authors made in rebuttal. However, the paper still seems to be far below the bar of the conference to me.
> >
> > For theoretical analyses, even if this is not an assumption of practical descriptors, such an assumption still makes the proof of theorem too straightforward, which significantly weakens the theoretical contributions.
> >
> > For experimental improvement, it is too tricky to look at relative improvement mentioned in the rebuttal.
> >
> > For model backbones, I suggest all experiments should be conducted again with modern backbones like ResNet-50 or ViTs. Otherwise, such experimental results are not really persuasive nowadays.
> >
> > Therefore, I decide to keep my score.

---

> > > ### Author Response · Authors · 2025-08-08
> > >
> > > We appreciate the reviewer’s comments. However, it is unclear why the proof is still deemed “too straightforward” and the theoretical contribution “significantly weakened.” The comment does not identify any error or logical flaw in the theorem or its proof. We would appreciate further clarification on the specific standards applied here.
> > >
> > > For backbone choice, we select architectures aligned with each dataset’s characteristics to ensure fair and effective evaluation. In particular, fMoW naturally fits large backbones like ResNet-50, and we have already provided corresponding results.

---

> ### Author Response · Authors · 2025-08-05
>
> Dear Reviewer aVTi,
>
> We greatly appreciate your time to review our paper and your comments.
>
> We noticed that we haven’t yet received your response and wanted to kindly inquire if there’s anything further we can do.
>
> We have made extensive clarifications, and provided additional experimental results as you suggested. We hope we have effectively addressed your concerns and clarified any potential misunderstandings.
>
> We are eager to hear your thoughts on the efforts we have made during the rebuttal period.
>
> Thank you once again for your review.

---

### Official Review · Reviewer_rCQN · 2025-07-02

**Clarity:** 2
**Significance:** 3
**Originality:** 3
**Rating:** 4
**Confidence:** 2

**Summary:**

This paper introduces Continuous Domain Generalization, extending domain generalization from discrete/temporal settings to arbitrary continuous descriptor spaces. The authors propose a Neural Lie Transport Operator (NeuralLTO) that learns to transport model parameters between domains while enforcing geometric continuity and algebraic consistency. They prove that optimal parameters form a low-dimensional manifold and include mechanisms for handling noisy/incomplete descriptors.

**Questions:**

1. How does NeuralLTO perform when P(Y∣X,z) changes drastically  (e.g., sim-to-real gaps)? Have you measured cycle-consistency error in such cases?

2. The method fixes several important hyperparameters: k=5 neighbors for local charts (Equation 9) and a 32-dimensional latent space for parameter embedding. How sensitive are the results to these choices? Specifically, can you provide sensitivity analysis showing performance.

**Ethical Concerns:**

["NO or VERY MINOR ethics concerns only"]

**Final Justification:**

My concerns have been addressed, and I kept my score.

**Limitations:**

Yes

**Quality:**

3

**Strengths And Weaknesses:**

**Strength:**

1. Generalizes beyond temporal domain shifts to multi-dimensional continuous spaces
2. Theorem 1 provides geometric justification for parameter manifold structure
3. Lie group theory provides elegant mathematical framework


**Weaknesses:**

1. Evaluation on toy or lightly filtered datasets. Experiments focus on 2-Moons with a synthetic “rotation” descriptor and a small Climate-fMoW split . These shifts resemble data augmentation more than uncontrolled, multi-factor drift, so real-world value is unclear.
2. In the middle of the proof of Theorem 1, the authors assume Jz θ(z) has rank d. This pre-condition is not listed up front, and real descriptors rarely satisfy it; the manifold guarantee may therefore not apply. Please put assumption Cleary visible.
3. Definition 2 demands closure, identity, associativity and invertibility, yet the NeuralLTO implementation is a neural approximation of the exponential map. However, the paper provides no experimental validation that these group properties actually hold in practice for the learned transport operator.
4. Each domain stores its own parameter vector θᵢ plus a private head inflating model size. Appendix A.4 claims overhead is “controlled” .

---

> ### Author Rebuttal · Authors · 2025-07-30
>
> Thanks for reviewing and acknowledging our work. Please find our response as follows:
> > W1. Evaluation on toy or lightly filtered datasets. Experiments focus on 2-Moons with a synthetic “rotation” descriptor and a small Climate-fMoW split . These shifts resemble data augmentation more than uncontrolled, multi-factor drift, so real-world value is unclear.
>
> 1) Our evaluation includes four large-scale real-world datasets with **uncontrolled and multi-factor domain shifts**:
>
> - fMoW involves complex shifts from factors as temperature, precipitation, humidity, and radiation;
> - ArXiv captures semantic drift across disciplines and publishers;
> - Yearbook spans decades of social, technological, and stylistic change;
> - Traffic reflects variation in POI distributions, urban function, and socio-economic activity.
>
> These shifts are **inherent, naturally occurring, and not based on data augmentation**. All datasets are widely recognized for real-world distribution shift: fMoW, ArXiv, and Yearbook are from paper in NeurIPS21 DB Track [55] (citation #121), one of the most comprehensive and widely adopted datasets for distribution shift. Traffic from paper in KDD19 [41](citation #647), and has been used in numerous urban analytics and spatiotemporal generalization studies.
>
> 2) The use of synthetic datasets (2-Moons, MNIST) follows standard ML practice, providing concise, interpretable ground-truth variation that contributes to rigorously evaluating the mathematical behavior of models. Importantly, existing methods all perform poorly, **confirming the non-triviality of the task even under such seemingly ‘toy’ settings**.
>
> > W2. In the middle of the proof of Theorem 1, the authors assume Jz θ(z) has rank d. This pre-condition is not listed up front, and real descriptors rarely satisfy it; the manifold guarantee may therefore not apply. Please put assumption Cleary visible.
> 1) We appreciate the comment. To clarify, **Theorem 1 does not assume the descriptors used in practice**. Rather, it expresses that a complete, non-degenerate descriptor space theoretically exists and characterizes the model parameter structure. Such a theoretical descriptor space is grounded in reality, as distribution shifts are ultimately governed by a finite set of latent factors.
> 2) Practical descriptors are imperfect. We fully acknowledge this, but  **practical descriptors serve as projections of the theoretical space**, so having partial descriptors is still better than having none. Moreover, the underlying manifold established in Section 4.1 **exists independently of practical descriptor quality**. Our method explicitly addresses the imperfect problem in Section 4.3. Experiments in real-world datasets and Section 5.3 further demonstrate that these partial descriptors are informative and effective.
> 3) We will revise the statement of Theorem 1 to clarify this point and remove ambiguity.
>
> > W3. Definition 2 demands closure, identity, associativity and invertibility, yet the NeuralLTO implementation is a neural approximation of the exponential map. However, the paper provides no experimental validation that these group properties actually hold in practice for the learned transport operator.
> 1) NeuralLTO is grounded in Lie group theory and is defined via exponential maps in Lie algebras, which theoretically satisfy group properties by definition.
> 2) To validate it, we get the $\theta$ for all test domains of 2-Moons, then conduct the following evaluations:
> - **Identity**: For each test descriptor $ z_i $, we apply $\theta(z_i) \xrightarrow{z_i \rightarrow z_i} \theta'(z_i)$ and evaluate the cosine similarity: $\cos(\theta(z_i), \theta'(z_i))$.
> - **Associativity**: We consider all triplet combinations $(z_i, z_j, z_k)$ from the test descriptors. For each triplet, we get two results: $\theta(z_i) \xrightarrow{z_i \rightarrow z_j} \theta'(z_j) \xrightarrow{z_j \rightarrow z_k} \theta'(z_k)$ and $\theta(z_i) \xrightarrow{z_i \rightarrow z_k} \theta''(z_k)$, then compute $\cos\left( \theta'(z_k),\ \theta''(z_k) \right)$
> - **Invertibility**: We consider all pairs $(z_i, z_j)$ from the test descriptors. For each pair, we apply $\theta(z_i) \xrightarrow{z_i \rightarrow z_j} \theta'(z_j) \xrightarrow{z_j \rightarrow z_i} \theta'(z_i)$, then compute $\cos\left( \theta(z_i),\ \theta'(z_i) \right)$.
> - **Closure** is inherently satisfied, as the transport operator always maps within the valid parameter space.
>
> Results are as follows:
> | Property| Cosine Similarity|
> |-|-|
> | Identity| 99.9%|
> | Associativity|99.1%|
> | Invertibility|98.3%|
>
> These results indicate that the learned operator satisfies the group axioms with high fidelity.
>
> We appreciate your constructive comments and will incorporate the above content into the camera-ready version.
>
> > W4. Each domain stores its own parameter vector $\theta_i$ plus a private head inflating model size. Appendix A.4 claims overhead is “controlled” .
> 1) Domain-specific parameters are confined to the final layers (e.g., lightweight MLPs), while the feature extractor (e.g., CNN backbone), which accounts for the vast majority of model parameters, is fully shared. This greatly limits the additional cost per domain.
> 2) After training, we can store only the parameter embedding vector $e$ of each domain, which further reduces cost.
> 3) The transport operator is also lightweight, just consisting of a few MLP layers.
>
> > Q1. How does NeuralLTO perform when P(Y∣X,z) changes drastically (e.g., sim-to-real gaps)? Have you measured cycle-consistency error in such cases?
> 1) We clarify that CDG assumes distribution continuous shifts, in line with the **continuum hypothesis** widely adopted in physics and engineering: every real-world macroscopic systems evolve smoothly over time, with their physical quantities modeled as continuous fields governed by differential equations.
> 2) What may appear as a "gap" reflects **coarse or sparse sampling** over an underlying continuous variation field. This is not a limitation for our method: as shown in Figures 5 left and 6, the training domains are intentionally designed to be **extremely sparse in the descriptor space**, yet the model generalizes smoothly and performs robustly across the full variation range.
>
> > Q2. Sensitivity Analysis
> 1) We have already conducted a sensitivity analysis in Appendix A.7 (Figure 8), varying the number of neighbors $k$ and the dimensionality of the parameter embedding.
> 2) The results show that the performance remains stable across a range of values, demonstrating that our method is not sensitive to these choices.

---

> > ### Comment · Reviewer_rCQN · 2025-08-05
> >
> > Thank you for the rebuttal and clarifications. My concerns have been addressed, and I’ll keep my score.

---

### Note · Authors · 2025-08-13

We thank all reviewers. Three reviewers, rCQN, sRq5, DmZF, express broadly consistent views: they recognize our core contributions of (i) extending DG to arbitrary continuous descriptor spaces (rCQN Strengths 1; DmZF Strengths 2), (ii) introducing Operator with geometric continuity + algebraic consistency (rCQN Strengths 2; DmZF Summary), and (iii) providing theoretical justification and mechanisms (rCQN Strengths 3; sRq5 Strengths 1).

During rebuttal, we addressed their main concerns with specific additions and clarifications:
- Theory assumption: Clarified Theorem 1 is a theoretical construct.
- Algebraic properties: Provided empirical verification.
- Motivation & PDE vs neural operator: Clarified the problem formulation and why PDE-based solvers are inapplicable.
- Backbone size & dataset realism: Added ResNet-50 on fMoW, showing consistent gains.
- Runtime on a larger backbone was reported.

**All three reviewers explicitly stated that their concerns had been well addressed** (rCQN: 'My concerns have been addressed'; sRq5: 'most of my concerns have been addressed'; DmZF: 'Most of my concerns and questions have been addressed').

One outlier—Reviewer aVTi raised three main concerns:

- Non-degeneracy assumption — We clarified that Theorem 1 is a theoretical construct, independent of practical descriptor perfection, and our method explicitly handles imperfect descriptors
- Performance gains — We demonstrated 12% average relative improvement over descriptor-agnostic baselines and 9% over SOTA approaches, which are considered significant in DG literature
- Baselines/backbones — We added stronger baselines, Fishr, and conducted new ResNet-50 experiments on fMoW, confirming consistent SOTA performance; the remaining datasets are not suitable for ResNet backbones due to their image scale.

In the final comment, reviewer aVTi argued the proof is “too straightforward,” rejected relative improvements, and requested all experiments be re-run regardless of the data scale. Notably, no fatal flaw or error was identified, yet Strong Reject (score 1) was maintained, which is an apparent mismatch between the textual rationale and extreme score.

To summarize, the majority view acknowledges novelty and confirms that most technical concerns were resolved. We respectfully hope the AC will weigh this consensus and our added evidence when forming the decision.

---

### Decision · Program_Chairs · 2025-09-17

**Decision:**

Accept (poster)

**Comment:**

The reviewers acknowledged the interest and novelty of the problem addressed, the elegant mathematical framework, the diverse application domains covered, and the motivation behind the approach. Nevertheless, they expressed concerns related to some aspects of the theoretical derivations, and some aspects of the empirical evaluation. The authors' feedback addressed the concerns of most of the reviewers, with the exception of Reviewer aVTi. Specifically, Reviewer aVTi expresses concerns about the relationship between Theorem 1 and practical descriptors, the performance gains reported by the authors, and the lack of experiments with modern backbones. The AC nonetheless believes that the authors' feedback satisfactorily addresses these points:
- Making assumptions in the proof of a theorem and not focusing on specific descriptors is standard, and the authors have clarified this in their answer;
- The performance improvements reported by the authors are typical of what can be found in the literature;
- Experiments with a ResNet-50 backbone were provided in the rebuttal. The AC acknowledges that this point could be further strengthened with, e.g., a ViT backbone. However, the current set of experiments already shows that the method is effective.

Altogether, the AC therefore believes that the positive aspects listed by the reviewers outweigh the negative ones, and that the paper is sufficient for acceptance. The authors are nonetheless strongly encouraged to incorporate elements of their answers in the final version of the paper.